# Genomic and structural insights into Jyvaskylavirus, the first giant virus isolated from Finland

Gabriel Magno de Freitas Almeida[1]*, Iker Arriaga[2], Bruna Luiza de Azevedo[3], Miika Leppänen[4], Jonatas S Abrahão[3], Julien Andreani[5,6], Davide Zabeo[7†], Janne J Ravantti[8], Nicola GA Abrescia[2,9]*, Lotta-Riina Sundberg[4]

[1]The Norwegian College of Fishery Science, Faculty of Biosciences, Fisheries and Economics, UiT - The Arctic University of Norway, Tromsø, Norway; [2]Structure and Cell Biology of Viruses Lab, CIC bioGUNE, Basque Research and Technology Alliance (BRTA), Derio, Spain; [3]Universidade Federal de Minas Gerais, Institute of Biological Sciences, Department of Microbiology, Belo Horizonte, Brazil; [4]University of Jyväskylä, Department of Biological and Environmental Science and Nanoscience Center, Jyväskylä, Finland; [5]Aix Marseille Univ, MEPHI, Marseille, France; [6]IHU-Méditerranée infection, Marseille, France; [7]Diamond Light Source, Harwell Science and Innovation Campus, Didcot, United Kingdom; [8]University of Helsinki, Molecular and Integrative Biosciences Research Programme, Helsinki, Finland; [9]Ikerbasque, Basque Foundation for Science, Bilbao, Spain

*For correspondence:
gabriel.d.almeida@uit.no
(GMdFA);
nabrescia@cicbiogune.es
(NGAA)

Present address: †Department of Chemistry and Molecular Biology, University of Gothenburg, Gothenburg, Sweden

Competing interest: The authors declare that no competing interests exist.

## eLife Assessment

This manuscript describes an **important** study of the giant virus Jyvaskylavirus. The characterisation presented is **compelling**. The work will be of interest to virologists working on giant viruses as well as those working with other members of the PRD1/Adenoviridae lineage.

**Abstract** Giant viruses of protists are a diverse and likely ubiquitous group of organisms. Here, we describe Jyvaskylavirus, the first giant virus isolated from Finland. This clade B marseillevirus was found in *Acanthamoeba castellanii* from a composting soil sample in Jyväskylä, Central Finland. Its genome shares similarities with other marseilleviruses. Helium ion microscopy and electron microscopy of infected cells unraveled stages of the Jyvaskylavirus life cycle. We reconstructed the Jyvaskylavirus particle to 6.3 Å resolution using cryo-electron microscopy. The ~2500 Å diameter virion displays structural similarities to other Marseilleviridae giant viruses. The capsid comprises of 9240 copies of the major capsid protein, encoded by open reading frame (ORF) 184, which possesses a double jellyroll fold arranged in trimers forming pseudo-hexameric capsomers. Below the capsid shell, the internal membrane vesicle encloses the genome. Through cross-structural and -sequence comparisons with other Marseilleviridae using AI-based software in model building and prediction, we elucidated ORF142 as the penton protein, which plugs the 12 vertices of the capsid. Five additional ORFs were identified, with models predicted and fitted into densities that either cap the capsomers externally or stabilize them internally. The isolation of Jyvaskylavirus suggests that these viruses may be widespread in the boreal environment and provide structural insights extendable to other marseilleviruses.

**eLife digest** Viruses are everywhere. Some viruses can cause illness in humans, which has led to people often viewing them as threats. But most viruses are harmless to humans. Many viruses target microbes instead of humans. These viruses likely play essential roles in maintaining a healthy balance in ecosystems by preventing overpopulation of their target microbes. Yet, scientists know very little about most viruses and their role in ecosystems.

Scientists have recently discovered a special group of giant viruses that target microscopic creatures called amoebas. The giant viruses that target them are much larger than most viruses and have some other unique features, such as having large genomes. A tough outer coating provides the structural reinforcement necessary to support their large size.

Most of the giant viruses identified so far have been discovered in Europe and South America. A few have been found in North Africa, India, Japan and Siberia. But scientists still do not know how many giant viruses there are and where else they exist around the world. They are also still learning about the structures that allow these giant viruses to be so large.

Almeida et al. describe the first giant virus ever discovered in Finland. In the experiments, the researchers mixed a type of amoeba called *Acanthamoeba castellanii* with environmental samples and monitored the amoebas for infection. The experiments revealed a giant amoeba-infecting virus in a compost sample. Almeida et al. named it Jyvaskylavirus after the city from which the compost sample came. The investigators then sequenced the virus' DNA and used a cutting-edge imaging tool called cryogenic electron microscopy and artificial intelligence to determine details of the viral structure.

This detailed structure of Jyvaskylavirus provided information about the structure of giant viruses and key structural proteins, which may also benefit scientists studying other kinds of viruses. The experiments confirm that giant viruses are a part of Finland's boreal forest ecosystem, extending the known range of these unusual viruses. More studies are needed to identify the full range of giant viruses worldwide and to understand their role in ecosystems.

## Introduction

Viruses defy paradigms of classical biology and are agents of change and innovation for biological research. In 2003 researchers were surprised by the description of the *Acanthamoeba polyphaga mimivirus* (APMV) (*Scola et al., 2003*). Hidden in plain sight for over a century of microbiology research, APMV and other giant viruses were evading discovery for three main reasons: their large size traps them in filters commonly used in virological works, their structure and size makes them visible by Gram staining, and their protist hosts are not as well studied as other host species (*Colson et al., 2017*; *Queiroz et al., 2022*). Common features of their double-stranded DNA genomes, similarity in a few core genes, relative independence of host transcription machinery, partial or complete cytoplasmic replication and formation of viral factories include giant viruses in the nucleo-cytoplasmic large DNA viruses (NCLDVs) group (*Iyer et al., 2001*; *Boyer et al., 2010*). Recently, the International Committee of Taxonomy of Viruses (ICTV) classified giant viruses as members of the phylum *Nucleocytoviricota* of the kingdom *Bamfordvirae* in the realm *Varidnaviria* (*Walker et al., 2019*; *Simmonds et al., 2023*).

Isolation efforts and metagenomic data in the last two decades have revealed that giant viruses are ubiquitous in the environment (*Moniruzzaman et al., 2020*; *Schulz et al., 2020*). Giant virus particles or their DNA have been found from the Antarctica to the Siberian permafrost, in deep-sea sediments and many other sources including urban environments and diverse sample types (*Andrade et al., 2018*; *Legendre et al., 2015*; *Bäckström et al., 2019*; *Colson et al., 2017*). To support the sharp rise in metagenomic studies, microbial isolation and characterization are welcomed, and needed for deeper understanding of these entities and the viral world. Isolation of giant viruses revealed the intriguing mosaicism of Marseillevirus, the complex structure and mysterious large genome of Pandoravirus, the remarkable structure and translational potential of Tupanvirus, and many other characteristics of these organisms that have changed our view on the concept of viruses and their evolution (*Boyer et al., 2009*; *Philippe et al., 2013*; *Abrahão et al., 2018*). It is expected that many new insights will emerge as more isolates are found (*Aherfi et al., 2016*; *Colson et al., 2017*). Increased effort in isolating giant viruses might also bring new discoveries and help in understanding their distribution and importance worldwide. Isolation of viruses allows for in-depth structural studies using their whole virions.

An increased number of icosahedral NCLVD, thanks to the advances in cryo-electron microscopy (cryo-EM), are now being targeted for structural analysis despite the challenges that their very large dimensions pose (>1500 Å diameter). Cryo-EM structures for African swine fever virus (ASFV), *Aureococcus anophagefferens* virus, *Cafeteria roenbergensis* virus, Faustovirus, Marseillevirus, Medusavirus, Pacmanvirus, Paramecium bursaria Chlorella virus 1, *Phaeocystis pouchetii* virus, and more recently for Melbournevirus have elucidated their complex architecture, major capsid protein (MCP) fold, and assembly organization (*Andreani et al., 2017*; *Andrés et al., 2020*; *Burton-Smith et al., 2021*; *Gann et al., 2020*; *Klose et al., 2016*; *Okamoto et al., 2018*; *Shao et al., 2022*; *Wang et al., 2019*; *Watanabe et al., 2022*; *Xiao et al., 2017*; *Yan et al., 2005*). Another example of the importance of the combination of isolation and structural studies is the description of FLiP, a missing link between ssDNA and dsDNA viruses in Finland (*Laanto et al., 2017*). However, FLiP is a bacteriophage and there have been no previous studies on the isolation of giant viruses in the Finnish boreal ecosystem or have their structural characterization been described so far.

Microbial ecology and virus-host interactions are still poorly studied in many environments, including boreal ecosystems. Here, we report the occurrence of giant viruses in Central Finland and the characterization of Jyvaskylavirus, a new virus belonging to the clade B of the Marseilleviridae family as determined by genome analysis. This virus was isolated from a composting soil sample in the city of Jyväskylä and represents the northernmost marseillevirus known to date. Other marseilleviruses from the northern hemisphere were found in France, India, Japan, Algeria, and Senegal only (*Sahmi-Bounsiar et al., 2021*), while closer giant viruses are either an uncharacterized Swedish cedratvirus (Lurbovirus) (*Kördel et al., 2021*) or a few microalgae-infecting mimivirus-like and phycodnaviruses-like isolates from Norway (*Castberg et al., 2002*; *Sandaa et al., 2001*; *Johannessen et al., 2015*). We used helium ion microscopy and transmission electron microscopy to visualize the early attachment events of the virus to its amoebal host (*Acanthamoeba castellanii*) and the infected cell. Using cryo-EM, we also determined the three-dimensional (3D) structure of Jyvaskylavirus at a resolution of 6.3 Å. Cross-structural and sequence comparison allowed us to identify five proteins that compose the capsid essential for assembly, with their corresponding models reliably placed into density.

Jyvaskylavirus description is the first step in unveiling the diversity of giant viruses from Finland and from the Nordic countries, exemplifying that these viruses are also present in the boreal ecosystems with a still unknown role for microbial ecology.

## Results

### Giant viruses are present in Finland

During the summer 2019 a preliminary isolation attempt of local amoebas and giant viruses using samples collected in Central Finland hinted at the presence of these viruses in Finnish samples (Appendix 1 and *Appendix 1—figure 1*). Although viral-like particles were seen, working with the local protist host presented challenges that initially hindered the virus characterization. We followed up this preliminary screening with a larger isolation effort using three reference host strains: *A. castellanii*, *A. polyphaga*, and *Vermamoeba vermiformis*. Ninety-six environmental samples collected in Central Finland were tested using the three different amoebal hosts. Of these samples, 10 had viral presence confirmed by the appearance of cytopathic effect (CPE) in cultures of *A. castellanii* followed by visualization of negative stained viral-like particles by electron microscopy (10.41% isolation success in *A. castellanii*). However, some of the samples belonged to the same sample group and viral morphology was similar between some of the isolates. Adjusting for sample type and unique viral morphologies found, we have then tested 53 unique sample types and found three distinct viral morphologies (5.66% isolation success in *A. castellanii*) (see *Supplementary file 1*).

Some of the CPE-positive samples contained mixed viral populations. Samples collected from experimental aquaria had a mixture of three different viral morphotypes with dimensions ranging from approximately 200 to 300 nm side-to-side, highlighting a previously unexplored diversity of giant viruses in this setting (*Figure 1A–C*). One sample from a recirculating aquaculture tank also possessed the diamond-shaped and loose-capsid morphologies seen in the aquaria samples. The high amount of nutrients in the fish aquaria or tanks might have enriched for protozoa and thus favored our giant virus isolation, in a pattern similar to the original methodology for isolating giant viruses (*Arslan et al.,*

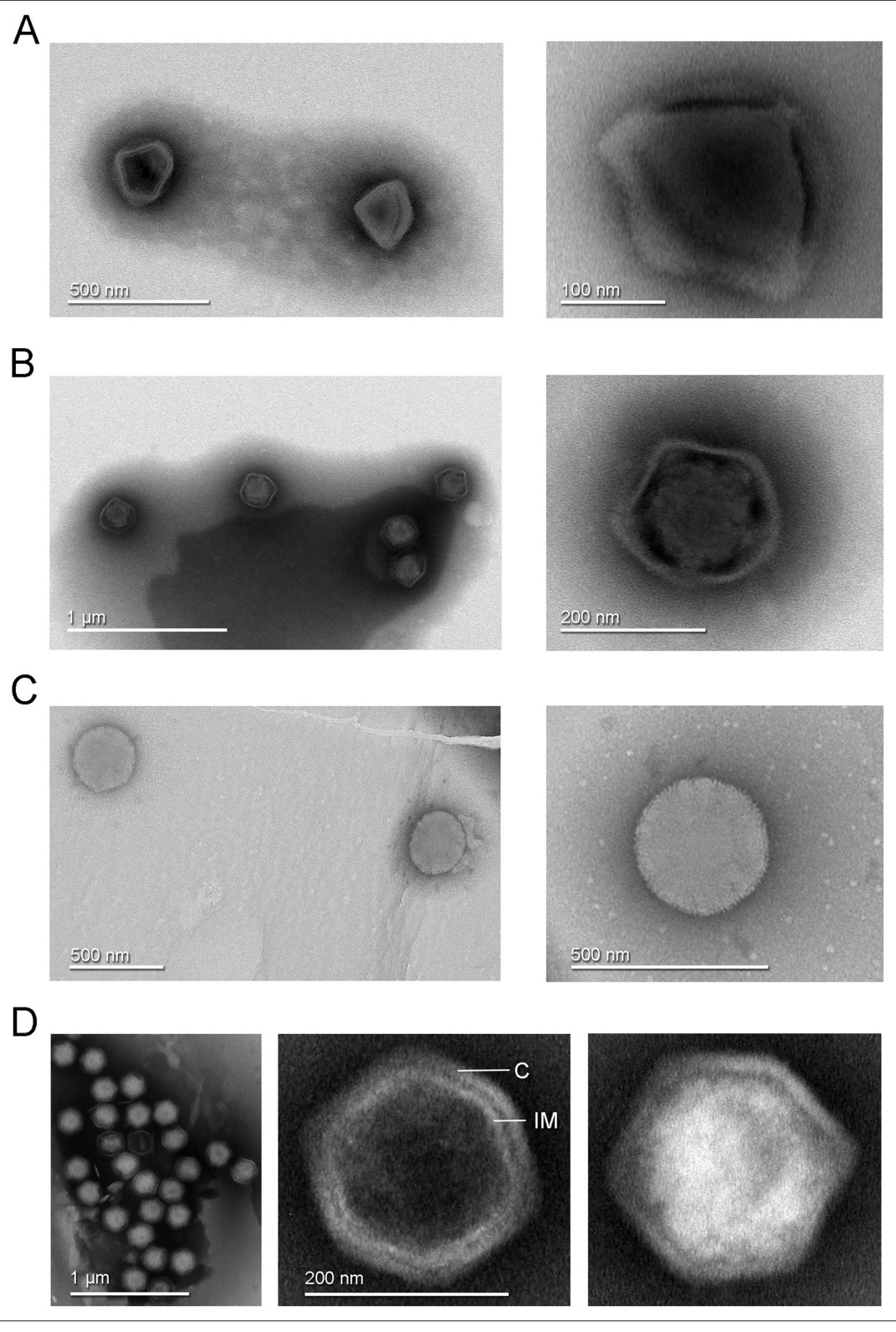

**Figure 1.** Transmission electron microscopy images of negative stained virus-like particles isolated from Finnish samples. (**A**) Left, diamond-shaped virions found in a Recirculating Aquaculture System (RAS) tank sample and in experimental aquarium samples. Right, enlarged view showing the virion interior with at least three attachment points to the external capsid. (**B**) Left, virions with loosely structured capsids found in a RAS tank sample and in experimental aquarium samples. Right, detailed view of one virion of this morphotype. (**C**) Left, round morphotype

*Figure 1 continued on next page*

*Figure 1 continued*

found in an experimental aquarium sample; right, enlarged view of a spherical-shaped virion. (**D**) Left, clusters of full and empty Jyvaskylavirus virions isolated from a composting soil sample. Center, an empty capsid showing the double-layered architecture of Jyvaskylavirus; C marks the capsid and IM marks the putative internal membrane. Right, a fully packaged Jyvaskylavirus virion.

*2011*). However, our attempts to isolate these viruses using limiting dilutions and more passages did not result in reliable purified samples, making it impossible to proceed with further characterization.

The isolate from sample 85 had a single morphotype of approximately 200 nm in size, displaying an icosahedral shape with particles that were mostly genome-filled and a few that were empty (*Figure 1D*). The empty particles possibly displayed an internal membrane beneath the capsid (*Figure 1D,* center).

This sample was collected from the municipal Waste Treatment Centre, Jyväskylä, Finland. The sample was in soil state and originated from a mixture of 70% garden waste, 15% woodchips, and 15% pretreated biowaste. Virus Ac-85 was chosen as a model to represent the first giant virus isolate from Finland and named Jyvaskylavirus as a homage to the city of its isolation. Jyvaskylavirus has a fast replication cycle in its *A. castellanii* host, only causes CPE in *A. polyphaga* or *V. vermiformis* in high concentrations likely due to cytotoxic effect of the virus preparation, is sensitive to chloroform treatment (10% for 10 min), and is stable for long periods (up to 109 days) even at 37°C (Appendix 1 and *Appendix 1—figure 2*). A fast replication cycle is a feature also shown for other marseilleviruses (*Boyer et al., 2009*; *Fabre et al., 2017*).

## Jyvaskylavirus belongs to the Marseilleviridae family

Jyvaskylavirus contains a 359,967 base pairs (bp) genome, with a 42.80% GC content and 388 predicted open reading frames (ORFs) coding for proteins with sizes that vary between 99 and 1525 amino acids. The positive DNA strand codes for 186 ORFs whereas the other 202 are in the negative strand (*Figure 2A*; *Supplementary file 2*). During genome annotation, some of the Nucleocytoviricota conserved proteins were detected, including the DNA polymerase family B (ORF177) and the A32-like packaging ATPase (ORF23). The most of Jyvaskylavirus genes (about 67%) code for uncharacterized proteins. The second major function category is the DNA replication, recombination, and repair genes, including three histone-like proteins (ORF215, 216, 320), a typical marseillevirus genome feature (*Figure 2B*; *Bryson et al., 2022*). Three new ORFans were detected (ORF264, 265, 289), representing genes that have no significant similarity with any other sequences from the database used in this analysis. ORF264, 265, and 289 codes for putative proteins with 108, 157, and 119 amino acids, respectively. We searched for translation-related genes and found three translation factors, including a translation initiation factor (ORF154), an elongation factor (ORF318), and a peptide chain release factor (ORF28). No tRNAs or aminoacyl tRNA synthetases genes were found. Furthermore, most of the BLASTp best hits for Jyvaskylavirus amino acid sequences matched Lausannevirus or Portmiou virus that are phylogenetically related to marseilleiviruses from lineage B. This observation can also be reinforced by the phylogeny based on DNA polymerase family B, which clusters the Jyvaskylavirus within *Marseilleviridae* family, together with other marseilleviruses from clade B (*Figure 2C*, *Appendix 1—figure 3A*). When analyzing the genome synteny of different marseilleviruses genomes, it is shown that Jyvaskylavirus presents similarity blocks comparable to those from clade B marseilleviruses (*Appendix 1—figure 3B*). Searching the Jyvaskylavirus MCP and DNA polymerase sequences in the MGnify database (*Richardson et al., 2023*) yields multiple hits with significantly low E-values (<1e-80), as expected from the apparent ubiquity of marseilleviruses. Of note was the detection of similar sequences in metagenomes and transcriptomes obtained from drinking water distribution systems of ground and surface waterworks in Central and Eastern Finland, evidencing that marseilleviruses are prevalent but still unexplored in this region (*Tiwari et al., 2021*).

## Jyvaskylavirus attachment to host cells and extracellular virion clusters by scanning helium ion microscopy

To visualize the interaction of Jyvaskylavirus virions with its *A. castellanii* host cells, we used a scanning helium ion microscope (HIM). This imaging technology has proven successful in the study of bacteriophages and their host bacteria in the past (*Leppänen et al., 2017*). We adapted the sample preparation methods to image cultures of *A. castellanii* infected with Jyvaskylavirus. Sample preparation

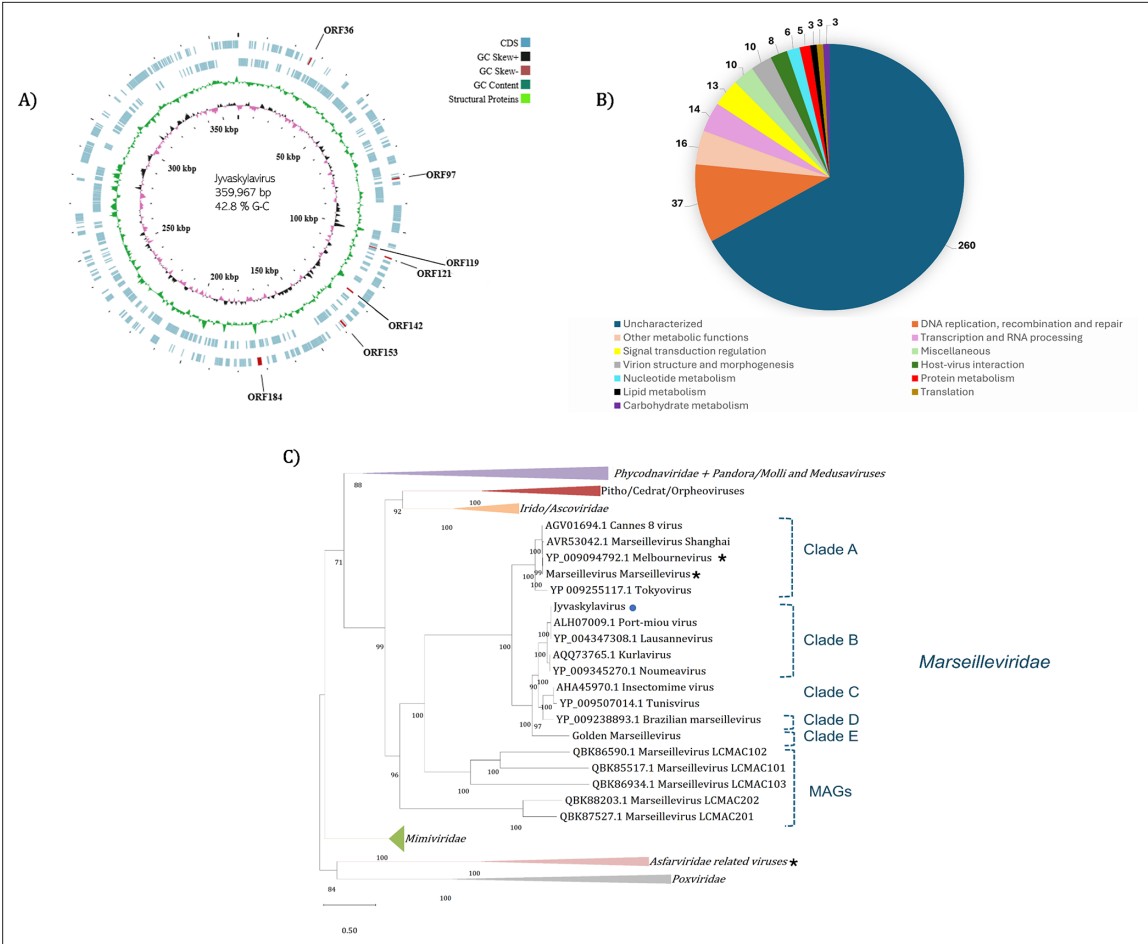

**Figure 2.** Jyvaskylavirus genomic data. (**A**) Representative map of Jyvaskylavirus genome features. The G-C content, G-C skew, and open reading frames (ORFs) distribution throughout the DNA sequence are coded by different ring colors as indicated in the color legend above. ORFs coding for the structural proteins mentioned in this paper are indicated by their ORF number. The outer blue ring represents the forward strand (positive sense) whereas the inner blue ring represents the reverse strand (negative sense). This illustrative genome map was constructed using CGView server (***Grant and Stothard, 2008***). (**B**) Number of Jyvaskylavirus proteins according to the function predicted during genome annotation; n=388. (**C**) Maximum-likelihood phylogenetic tree based on DNA polymerase family B amino acid sequences from different nucleocytoviruses. The Jyvaskylavirus sequence is indicated by a blue circle. An asterisk (*) marks close viruses with structures obtained by cryo-electron microscopy (cryo-EM). The alignment was performed with MUSCLE and the maximum-likelihood tree was reconstructed using IQtree software using ultrafast bootstrap (1000 replicates). The best-fit model selected using ModelFinder (implemented in IQtree) was VT+F+ R5. Scale bar indicates the number of substitutions per site.

was made by allowing cells colonize silicon chips treated with poly-l-lysine followed by infection and fixation, avoiding any need for cell scraping and pelleting for successfully imaging the amoeba cells and Jyvaskylavirus virions (***Figure 3***).

By HIM imaging *A. castellanii* cells appear as mainly oval-shaped with spined structures (acanthopodia) protruding from the cell surface; furthermore, a cell with an elongated shape was captured likely in the process of cell division (***Figure 3A***). No cysts were seen in these samples. At a closer range, cell surfaces appear rugose, full of crevices, and contains virions attached to it (***Figure 3B***). One of strategies for marseillevirus entry is the triggering of an endosomal-stimulated pathway (***Arantes et al., 2016***). We captured a cell with virions attached in regions showing invaginations of the cell membrane, probably indicating the process of endocytosis (***Figure 3C***). These craters, surrounded by walls with varying degrees of steepness, differ in size although the estimated diameter to accommodate at least one virion is about 300 nm (***Figure 3C***, inset). Alongside intact cells with attached virions, we also observed a broken cell nearby, with most of its cell contents released (***Figure 3D***, center). Extracellular vesicles larger than 500 nm in size containing multiple virions inside were visualized close to burst cells (***Figure 3D***, left inset). These virion clusters are important for marseilleviruses, which are

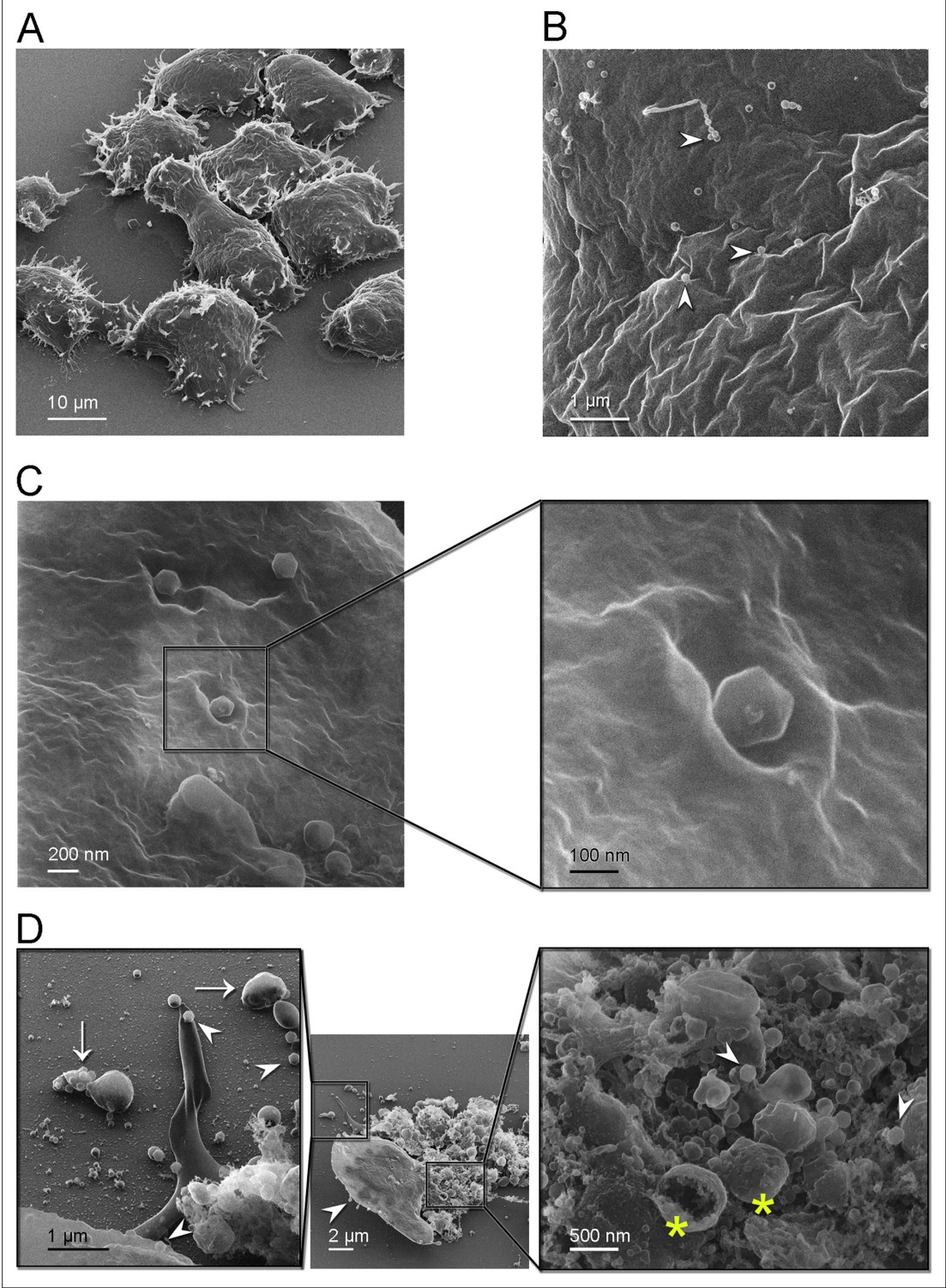

**Figure 3.** Helium ion microscopy images of Jyvaskylavirus attachment to *A. castellanii* cells. (**A**) *A. castellanii* cells with spined structures (acanthopodia); elongated cell at the center. (**B**) Details of a cell containing several viral particles on its surface (white arrowheads mark virions). (**C**) Icosahedrally shaped virions near cell surface invaginations appearing as craters; inset, details of a virion likely starting the infection process through endocytosis. (**D**) Center, one cell containing viruses on its surface (white arrowhead) near a burst cell displayed with its ruptured content. Right inset, details of the burst cell content, showing several vesicles (yellow asterisks) and viruses (white arrowheads). Left inset, clusters of virions inside extracellular vesicles indicated by white arrows and individual virions by white arrowheads.

smaller than other giant viruses, to reach the required threshold for triggering amoebal phagocytosis and thus initiating their infection cycle through this entry mechanism (*Arantes et al., 2016*). Details of the ruptured cell contents, revealing the presence of large internal spherical vesicles and intracellular virions, are shown in *Figure 3D* (right inset). Single virions were also observed on the substrate, suggesting that they could attach to foraging cells.

## Jyvaskylavirus forms intracellular vesicles and viral factories inside *A. castellanii* host cells

Intracellular details of the replication cycle of Jyvaskylavirus were imaged by transmission electron microscopy using ultrathin sectioning of cells undergoing CPE at 24 hr post-infection (hpi). At 24 hpi, the cells are overtaken by virus production, with intracellular vesicles of varying dimensions (e.g. 1–2 µm), which contain already apparent virions (*Figure 4A*). Large viral factories are observed, with some occupying almost the entire cell area (*Figure 4B*). The nucleus and other cellular components, such as mitochondria, can be clearly distinguished within the cell sections (*Figure 4A–C*). Several individual virions also populated the cytoplasm. One imaged vesicle is juxtaposed to the nuclear membrane and contains several assembled virions with clear icosahedral shape. These virion-rich intracellular vesicles are likely the source of the extracellular ones imaged by HIM in *Figure 3D* (left inset). Membrane-related structures located near or nearly attached to the luminal side of the large vesicle are discernible (*Figure 4C and D*). These membrane-derived assembled structures might serve during virus morphogenesis (*Figure 4D*). Inspection of the interior of viral factories unravels distinct stages of particle formation (*Figure 4E*). A putative assembly path can be extrapolated from the analysis of images of particles, which transition from a half-assembled icosahedron to a particle with an open vertex through the orderly accumulation of capsid proteins (*Figure 4F*). This open vertex is potentially used for genome packaging, and it is subsequently closed by the plugging of peripentonal and penton proteins.

## Jyvaskylavirus 3D architecture

Jyvaskylavirus icosahedral virion, determined to 6.3 Å resolution as judged by the gold-standard Fourier shell correlation, possesses a diameter of 2516 Å (vertex-to-vertex) and a triangulation number $T=309$ ($h=7$, $k=13$) (*Figure 5A*, *Appendix 1—figure 4*, and *Appendix 1—table 1*). The protein capsid shell, approximately 120 Å thick, can be geometrically represented by trisymmetrons and pentasymmetrons as similarly done with other NCLDVs (*Figure 5B*; *Sinkovits and Baker, 2010*). The trisymmetron and pentasymmetron comprises 136 and 30 pseudo-hexameric capsomers, respectively, along with one penton complex, containing five copies of the penton protein (*Figure 5B and C*). Jyvaskylavirus icosahedral asymmetric unit (IAU) is composed of 51 pseudo-hexameric capsomers plus 1/3 of the capsomer sitting on the icosahedral threefold axis (*Figure 5C*). Both the capsid organization and virion size are similar to those of other marseilleviruses, such as Melbournevirus and Tokyovirus. Pacmanvirus, considered to be at the crossroads between Asfarviridae and Faustoviruses, also possesses the same $T$ number (309) and a comparable diameter to Jyvaskylavirus. In contrast, other giant viruses, such as ASFV, representative of the Asfarviridae family, have a $T$ number of 277 and a diameter of approximately 2100 Å, while PBCV-1, a member of the *Phycodnaviridae* family, has a $T$ number of 169 and an average diameter of 1900 Å. All of the abovementioned viruses have been shown to possess an MCP with a vertical double jellyroll (DJR) fold that composes the capsid shell, along with an internal membrane bilayer. Minor capsid proteins have been identified and structurally modeled for the smaller virions ASFV and PBCV-1 (*Wang et al., 2019*; *Shao et al., 2022*).

Beneath the Jyvaskylavirus capsid, the membrane vesicle follows icosahedral symmetry, although it also displays a high degree of sphericity (*Figure 5A*, left). This internal membrane vesicle encloses the genome. Some particles, excluded during 2D classification, showed heterogeneous membrane morphologies indicating their structural fragility. At the fivefold axis between the capsid and the membrane, there is relatively weak density suggesting the presence of additional proteins (see below), along with a clear bulging of the membrane vesicle at the fivefold (radius of curvature of ~275 Å) (*Figure 5A*, left). To the best of our knowledge, this bulging has only been clearly observed in the closely related Melbournevirus and Tokyovirus.

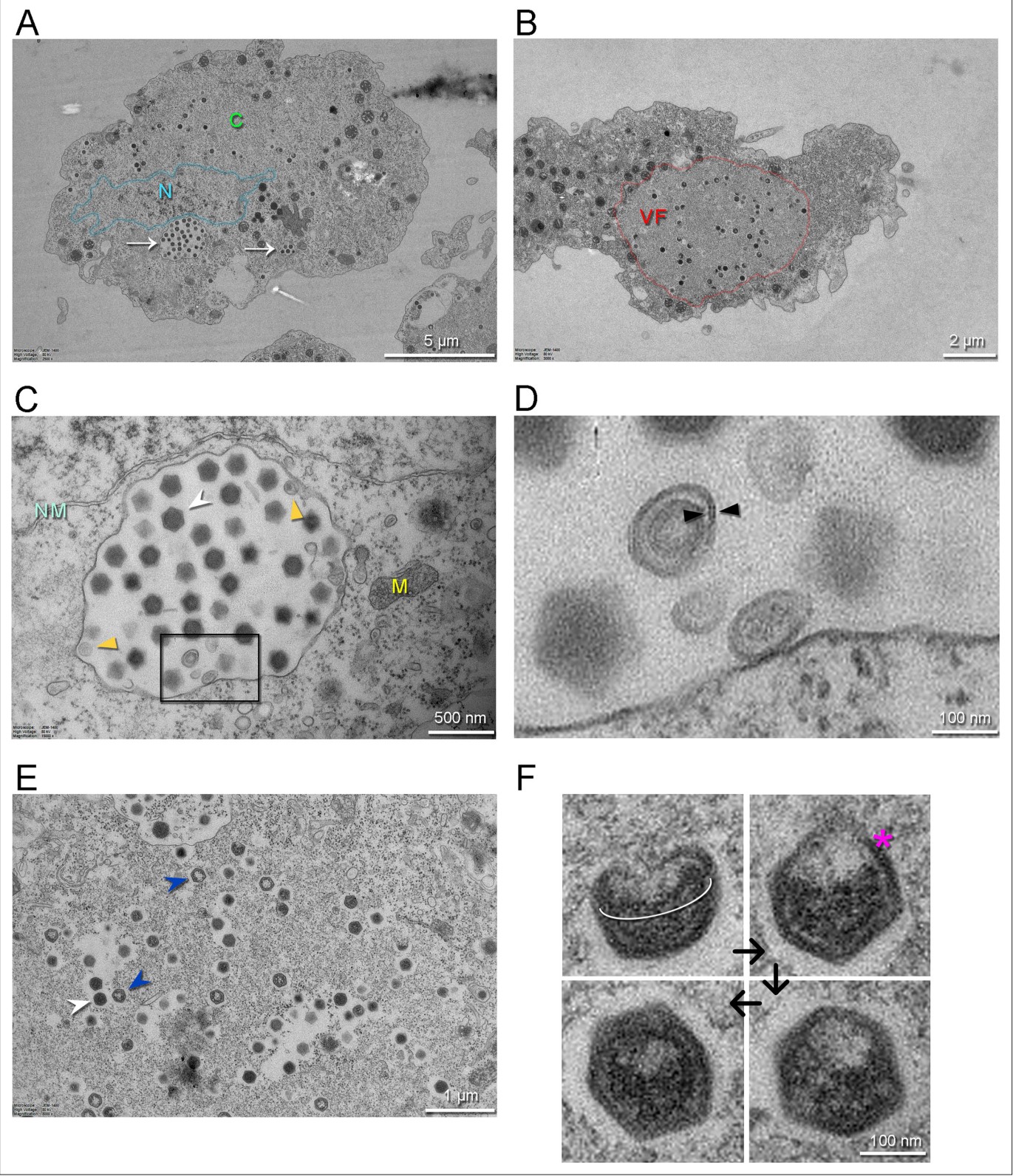

**Figure 4.** Transmission electron microscopy images of thin sections of *A. castellanii* cells infected by Jyvaskylavirus. (**A**) Infected cell containing viruses spread over its cytoplasm marked by C (green) and with intracellular vesicles filled with viruses indicated by white arrows. The nucleus, whose boundary is highlighted in semitransparent cyan, is indicated by the letter N (cyan). (**B**) One infected cell with a large viral factory (VF, red) in its cytoplasm. (**C**) View of an intracellular vesicle with icosahedral genome-filled virions as marked by a white arrowhead; membrane-related structures nearby the

*Figure 4 continued on next page*

Figure 4 continued

vesicle interior are marked by dark-yellow triangles. NM (light cyan) and M (yellow) mark the nuclear membrane and the mitochondria, respectively. (D) Enlarged view of the region marked by the black rectangle in (C) showing possible membrane-related structure juxtaposed to or detached from the vesicle interior; black triangles possibly indicated a forming membrane vesicle. (E) Details of virions in different stages of maturation inside the viral factory; DNA-full particles in white arrowheads, empty particles in blue arrowheads. (F) Putative stages of virion assembly (indicated by the black arrows) as derived from the inspection of distinct particles in E and other cellular sections. The white elliptical line highlights a capsid aperture, while the red asterisk indicates an assembling capsid; in the remaining virion images, the capsid appears more assembled.

## Jyvaskylavirus structural proteins composing the capsid

At 6.3 Å resolution alpha-helical secondary structural motifs are identifiable while the separation of β-strands became clearer beyond 5 Å. Our density corresponding to the capsomer displays unequivocally pseudo-hexameric morphology with the characteristic footprint of trimers formed by vertical DJR seen in other viruses of the kingdom *Bamfordviriae* (*Andrés et al., 2020*; *Ravantti et al., 2020*; *Simmonds et al., 2023*). A cross-section of the density shows the β-barrel walls, further supporting that the MCP possesses a vertical DJR fold (*Figure 5C*, inset). During genome annotation based on sequence homology with other marseilleviruses, ORF184 was identified as a potential MCP. We submitted ORF184 to AlphaFold3, which predicted a model with a DJR fold (*Figure 5D*; *Abramson et al., 2024*). The fitting of this model into density leaves no doubts about ORF184 being the MCP and a total of 154 copies of the MCP compose the IAU (*Figure 5C and E*).

Then, we structurally identified the penton protein of Jyvaskylavirus as ORF142 by using the latest version of ModelAngelo software with the hidden Markov model (HMM) search procedure against the publicly available cryo-EM block-reconstructed capsid densities of the Melbournevirus, a clade A marseillevirus, at ~3.5 Å resolution (EMD-37188, 37189, 37190) (for details, see Materials and methods and *Appendix 1—figure 5*; *Jamali et al., 2024*). This identification was based on the hypothesis of structural conservation among conserved capsid components and partial protein sequence conservation among members of the *Marseilleviridae*. Model prediction of the ORF142 in AlphaFold3 produced a compact eight stranded β-barrel (BIDG - CHEF) typical of a jellyroll, elaborated by extended loops between strands DC and EF and a long C-terminal tail (*Figure 6A*, left) (*Jumper et al., 2021*). The jellyroll core of the penton proteins showed a reasonable fit into the corresponding Jyvaskylavirus densities at 6.3 Å resolution (*Figures 5E and 6A*).

Additional density was observed atop the pseudo-hexameric capsomers, capping the central region formed by the ORF184 jellyroll towers and likely stabilizing the trimer from the exterior. Using the same methodological strategy as for the penton protein, we identified ORF121 as this cap protein, which possesses a β-barrel fold and forms a trimer, with its N-terminal ends inserting into the crevice formed by the MCP jellyroll towers (*Figure 6B*). Beneath the capsid shell, weaker density is visible at varying distances from the virus center and at different locations within the trisymmetron (*Figures 6B and 7A*, top left). Their positioning resembles that seen in Tokyovirus and Melbournevirus, which have been linked to pentasymmetron components (PCs) and minor capsid proteins (mCPs) with proposed roles in scaffolding, cementing, and zippering (*Chihara et al., 2022*; *Burton-Smith et al., 2021*). However, no information is currently available regarding their corresponding ORFs and/or 3D models.

Using the abovementioned approach (*Appendix 1—figure 5*), we identified four additional Jyvaskylavirus proteins - ORF36, ORF97, ORF153, and ORF119 - located beneath the pentasymmetron capsomers. We predicted their 3D models using AlphaFold3 and positioned them within the corresponding higher-resolution Melbournevirus densities, where the core of the different molecules generally fit well within the constraints of the map (*Appendix 1—figures 6–8*). We then placed all the newly identified and predicted Jyvaskylavirus 3D models into our cryo-EM density map at 6.3 Å resolution (*Figure 7*). Intriguingly, ORF119 was also identified as the protein that runs along the edge of the trisymmetron facets, acting as a glue between adjacent capsomers belonging to two different trisymmetrons (*Figure 7B* and *Appendix 1—figure 8*). The relative orientation of the ORF119 molecule beneath the pentasymmetron and along the edges of two trisymmetrons is about 90 degrees. However, not all the visible density beneath the capsid in the deposited Melbournevirus block-based reconstructed maps could be accounted for by the fitted structures. Interestingly, all AlphaFold3 predicted ORFs presented unordered regions, mostly at the N- and C-terminal ends, with lengths varying depending on the protein and their location. While we did not attempt further remodeling of these flexible regions into Melbournevirus densities, a more complete representation

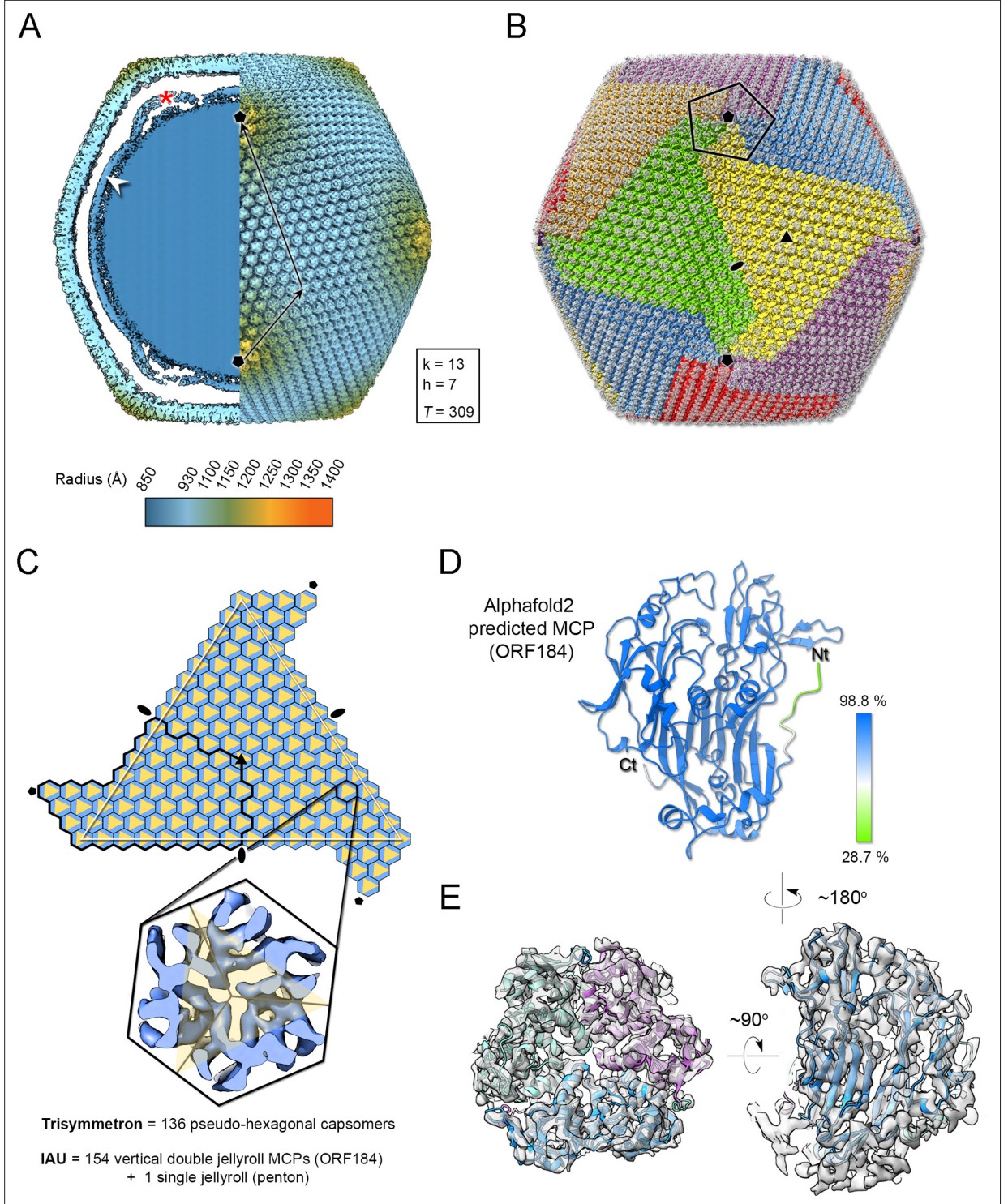

**Figure 5.** Jyvaskylavirus cryo-electron microscopy (cryo-EM) reconstruction. (**A**) Left, a central slab of the isosurface of the 3D density map, downsampled to a pixel size of 5.36 Å, low-pass filtered to 15 Å, and normalized, showing the interior of the virion color-coded by radius as from key. The white arrowhead marks the membrane bilayer, while the red asterisk highlights the membrane bulging beneath the fivefold vertex. Right, isosurface of half the virion with black arrows indicating the triangulation indices $h$=7, $k$=13 ($T$=309). A threshold level of 0.6 was used in ChimeraX to render both views (**Pettersen et al., 2004**). (**B**) Representation of the virion using trisymmetron geometry, with each trisymmetron color-coded differently, and a pentasymmetron marked by a pentagonal black line. (**C**) A schematic of a virus facet with a trisymmetron marked with a white triangle with three icosahedral asymmetric units (IAU), one of which is marked by a thick black line. The pseudo-hexameric morphology displayed by a capsomer is represented by a hexagon colored in light blue, while the true trimeric state of the major capsid protein (MCP) is depicted as a yellow triangle. To

*Figure 5 continued on next page*

*Figure 5 continued*

build the IAU (excluding the penton protein), 51 pseudo-hexameric capsomers and one-third of the capsomer located at the threefold symmetry axis are required, resulting in a total of 154 MCPs forming the IAU. The inset shows a cut-through of the density along the threefold axis of a capsomer. (**D**) AlphaFold3 prediction of the MCP ORF184 shows, with very high confidence, that the fold adopted by the ORF is a vertical double jellyroll. (**E**) Trimeric model of the capsomers rigid body fitted into the original cryo-EM density and rendered in ChimeraX; left, viewed along the trimer fold axis and on the right, viewed orthogonally to it. The three copies of the MCP are represented in cartoon and colored in green, light magenta, and light blue, while the corresponding density is shown in white transparent surface.

of Melbournevirus is achievable. To this end, we identified the corresponding Jyvaskylavirus ORFs in Melbournevirus through sequence comparison with Melbournevirus isolate 1 (NCBI Reference Sequence: NC_025412.1) (*Appendix 1—table 2*). However, when the identified Jyvaskylavirus ORF sequences were analyzed using BLASTp without restricting the search to the Melbournevirus reference, many hits were observed in other giant viruses, primarily marseillevirus. Remarkably, some of these hits scored higher than those for Melbournevirus, supporting the presence of homologous proteins in these viruses (*Appendix 1—table 3*). Further, a distinctive feature of the internal membrane vesicle is the bulging of the lipid bilayer at the fivefold equivalent to that found in Tokyovirus (*Figure 5A*, left and *Figure 6A*; *Chihara et al., 2022*). The identified PCs crown this region, and it is plausible that, along with yet unidentified proteins, they act as effectors of the bulging by tethering the membrane. The radius of the bulging is likely related to the extent of their localization beneath the pentasymmetron.

## Discussion

Jyvaskylavirus is the first characterized giant virus from Finland. So far the only other isolated giant viruses from the Nordic countries are the still uncharacterized Lurbovirus from Sweden and a collection of viruses capable of infecting microalgae isolated in Southern Norway (*Kördel et al., 2021Kördel et al., 2021*; *Sandaa et al., 2001*; *Johannessen et al., 2015*). There is genomic evidence of NCLDV presence in the Greenland ice sheet and of a high diversity of giant viruses, including the detection of marseilleviruses, in the Loki's Castle deep-sea vent located in the Mid-Atlantic Ridge between Iceland and Svalbard (*Perini et al., 2024*; *Bäckström et al., 2019*). However, no virus was isolated in these studies. Jyvaskylavirus belongs to clade B of the Marseilleviridae family, so far making it the northernmost known member of the family. The first marseillevirus was isolated from France in 2007 (*Boyer et al., 2009*). Now there are more than 60 isolates known, obtained from varied sample sources in seven countries over five continents (*Sahmi-Bounsiar et al., 2021*). Our genetic analysis showed that the Loki's Castle marseilleviruses group together and that they are not close to Jyvaskylavirus, probably forming unique clusters among themselves (*Appendix 1—figure 3*). Jyvaskylavirus is also unique for being the first found from a composting sample, demonstrating that these giant viruses are found in water and soil samples worldwide.

For the characterization of Jyvaskylavirus, we integrated complementary imaging techniques, including the use of HIM for imaging the virus and its *A. castellanii* host. This demonstrates the applicability of this technique to giant viruses and sets the methods for sample preparation, opening new ways for the study of giant viruses and interactions with their host.

Ultrathin TEM sectioning enabled us to capture snapshots of a putative assembly pathway, unraveling the progressive formation of the capsid. Single-particle cryo-EM provided critical insights into the structure of Jyvaskylavirus and its evolutionary relationship with other viruses. The 3D reconstructed cryo-EM density at 6.3 Å resolution clearly recapitulates the architecture ($T$=309; ~2500 Å vertex-to-vertex diameter) and capsid protein organization observed in Melbournevirus and Tokyovirus, other members of the *Marseilleviridae* family (*Burton-Smith et al., 2021*; *Chihara et al., 2022*). Although it is not possible to distinguish individual β-strands at this resolution, with the aid of Alpha-Fold3, we unequivocally identified and placed the predicted ORF184 MCP vertical DJR fold model into density. Additionally, we identified and predicted the fold of the penton protein encoded by ORF142 to adopt a vertical jellyroll topology. This model nicely fitted the corresponding density, plugging the center of the pentasymmetrons. There is structural conservation of the penton protein fold with that of other viruses such as penton proteins P31 in bacteriophage PRD1 or VP9 in archaeal virus HCIV-1 of the *Bamfordviria* and *Helvetiavirae* kingdoms, possessing vertical double and single jellyroll MCPs, respectively (*Abrescia et al., 2004*; *Santos-Pérez et al., 2019*). However, in ORF142, the CHEF

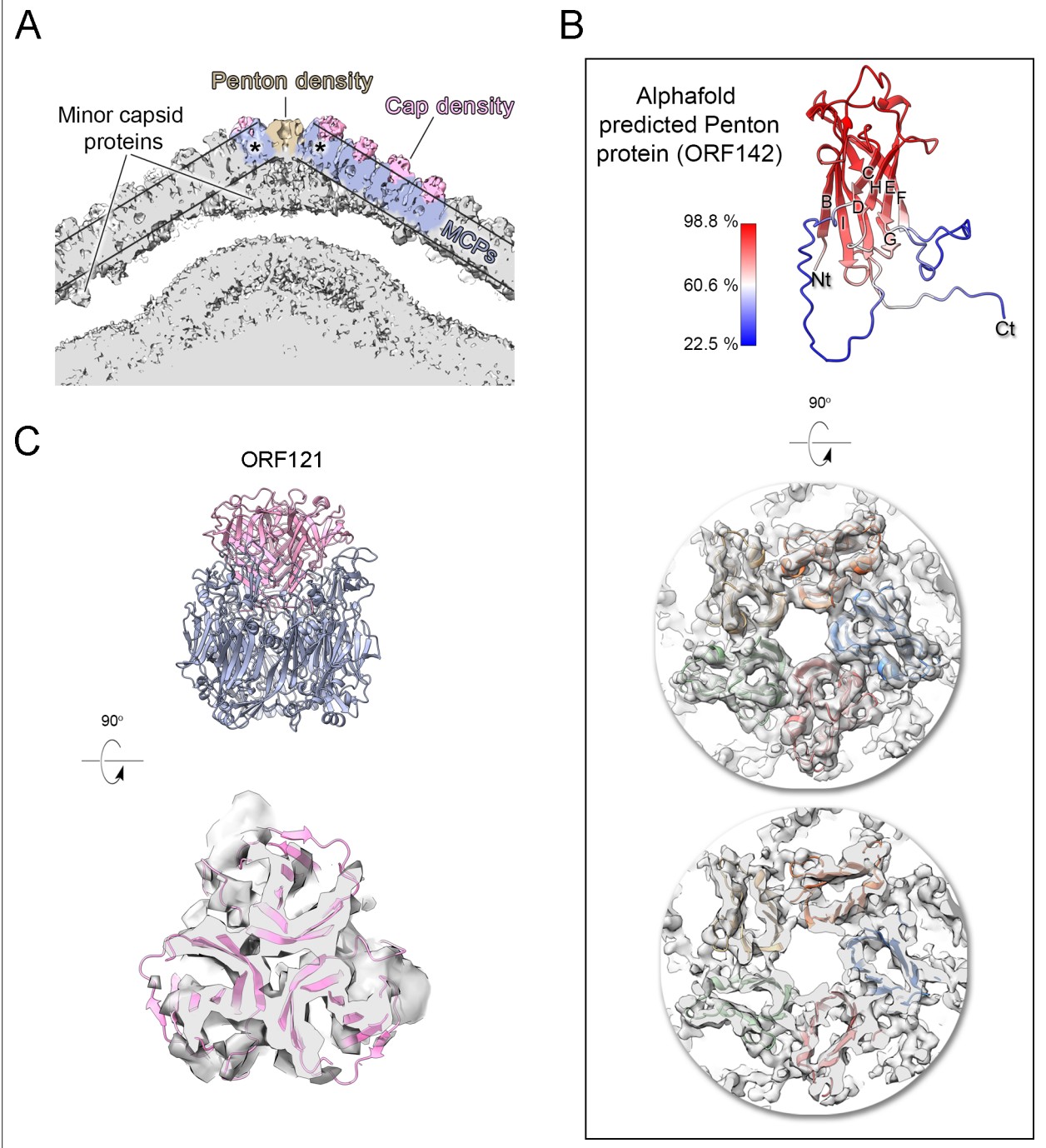

**Figure 6.** Jyvaskylavirus penton and capsomer-cap proteins. (**A**) Cut-through view of the virion cryo-electron microscopy (cryo-EM) density (binned twice from original size) at the fivefold axis rendered in gray in ChimeraX, with some regions of the capsid shell colored to highlight the penton density (light brown), the major capsid protein densities (slate-blue) and within the parallel outlined black lines and the cap densities (pink), other regions beneath the major capsid proteins (MCPs) and penton correspond to different minor capsid proteins; the black asterisks mark the peripentonal capsomers. (**B**) Top, a cartoon representation of the predicted model of the penton protein, with strands labeled BIDG/CHEF and color-coded by confidence level. Center and bottom, penton complex fitted into the density, shown from the top and as a cut-through, respectively. (**C**) Top, atomic model of the trimeric cap in pink cartoon, composed of three copies of ORF121 with a β-barrel fold inserted on top of the pseudo-hexameric capsomer, shown in slate-blue cartoon. Bottom, cut-through view of the cap model fitted into the original density, Gaussian filtered, and rendered as semitransparent gray in ChimeraX.

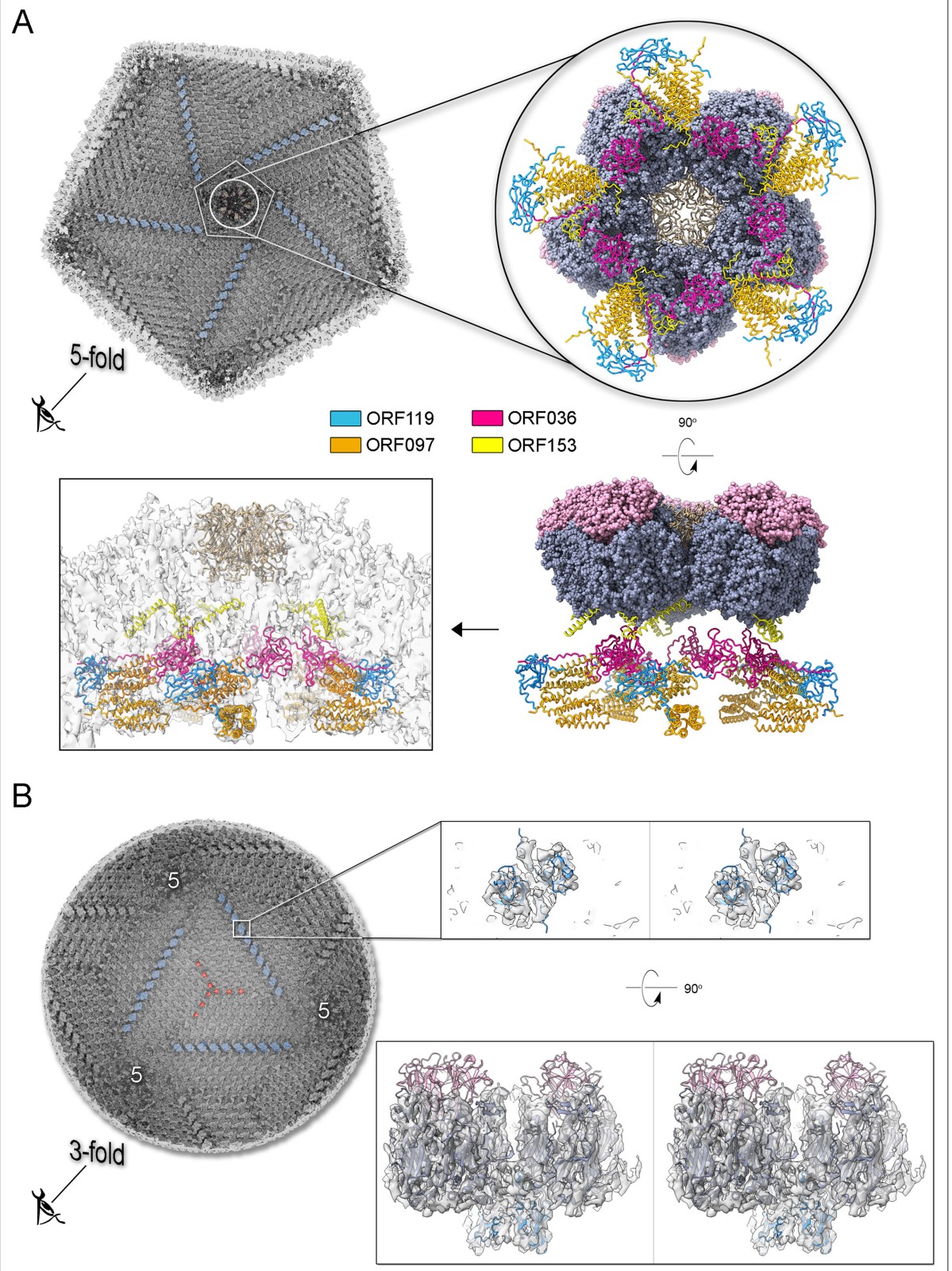

**Figure 7.** Jyvaskylavirus pentasymmetron protein components and trisymmetron facet glueing protein. (**A**) Top left, view of the cryo-electron microscopy (cryo-EM) density of Jyvaskylavirus (binned four times from the original size and Gaussian filtered) as seen from within the virion along the fivefold icosahedral symmetry axis. The white pentameric line marks the pentasymmetron region from below, while the white circle highlights the density corresponding to some identified proteins. Further densities corresponding to cementing, zippering, and lattice scaffolding proteins are also visible,

*Figure 7 continued*

with one of them colored in dodger-blue (see also B). The enlarged inset (top right and below) shows the spatial organization of the four identified open reading frames (ORFs) and modeled using AlphaFold3, represented as cartoon tubes and colored as per the legend. The penton proteins are colored light brown, while peripentonal capsomers and capsomer-cap proteins are shown as space-filled atoms and colored as slate-blue and pink, respectively. Left bottom, the different protein components fitted into the original 6.3 Å resolution Jyvaskylavirus cryo-EM density (white semitransparent) binned to 2.68 Å/pix and rendered in ChimeraX; major capsid proteins (MCPs) and capsomer-cap proteins have been omitted for clarity (for fitting metric, see Materials and methods). (**B**) Left, view of the cryo-EM density of Jyvaskylavirus, as shown in the top left panel of (**A**), but along the threefold icosahedral symmetry axis. A cementing protein, colored red, runs through the capsomers; however, the corresponding ORF has not been identified. Densities colored dodger-blue, which glue the capsomers across two trisymmetrons, correspond to ORF119, as shown in the large inset on the right. In this inset, stereo view of dimers of ORF119 are depicted as dodger-blue cartoons fitted into the original cryo-EM density, Gaussian filtered, and rendered as semitransparent gray in ChimeraX. At the bottom right, a 90 degree stereo view shows the spatial arrangement between two capsomers (navy blue), with the cap protein ORF121 (pink cartoon) positioned on top, and ORF119 fitted into the density at the base of the adjacent MCPs.

strands are predicted to be tilted relative to the BIDG strands, with an estimated angle of approximately 60 degrees based on visual inspection (*Appendix 1—figure 9*). This type of penton complex acts like a plug-and-play system, able to incorporate various host-recognizing vertex proteins. These proteins are interchangeable and adapt to environmental evolutionary pressures (*Gil-Carton et al., 2015*). The clear presence of a network of scaffolding, cementing, and minor proteins beneath, previously observed in ASFV and Tokyovirus, but also in smaller viruses such as PBCV-1, adenovirus, and PRD1, reiterates the universal requirement for the assembly of larger virus of an increased number of ancillary proteins tethering the MCPs at the vertices, within the facets and along the facets (*Abrescia et al., 2004*; *Burton-Smith et al., 2021*; *Liu et al., 2010*; *Wang et al., 2019*; *Chihara et al., 2022*; *Shao et al., 2022*). In this study, through cross-structural and sequence comparisons with available Melbournevirus block-based reconstructed densities and combination of AI-based modeling software (*Zhu et al., 2018*; *Burton-Smith et al., 2021*; *Jamali et al., 2024*), we were able to elucidate the identity and the fold of five additional viral capsid components in Jyvaskylavirus: ORF36, 97, 119, 153, and 121 (and the corresponding ORFs in Melbournevirus). The former four are located beneath the pentasymmetron while ORF121 is the protein capping on the top the capsomers and it shares its fold with marseillevirus Noumeavirus NMV_189 protein (PDB ID 7QRR) (e.g. rmsd 1.5 Å, 146 Cα aligned) and virophage Zamilion vitis protein Zav19 (PDB ID 7QRJ).

Remarkably, two ORF119 molecules that form a homodimer are also used to glue at their bases capsomers located at the edge of adjacent trisymmetrons. The orientation of the ORF119 molecule below the pentasymmetron differs from that along the edge of the trisymmetron, indicating that its binding mode depends on the structural environment. The reuse of the same protein in different regions of the facet for multiple functions highlights ORFs optimization and genetic parsimony. Particularly intriguing is that while AlphaFold3 predicts the core of most identified proteins, satisfying the constraints of the map, some regions are poorly ordered. One clear case is ORF153 which is predicted to possess about 40% of the residues unstructured. We propose that these regions together with flexible terminal ends are not merely a limitation of AlphaFold3's predictive capabilities but rather reflect specific functional characteristics of these proteins, which may fold and adapt through protein partnering. Additionally, the observation that the identified Jyvaskylavirus minor capsid protein sequences are shared across other marseillaviruses supports their essential structural and stabilizing roles in these viruses. It has been shown that the conserved flexibility and function of intrinsically disordered proteins, despite their often-fast rate and tolerance for mutations, plays critical physiological roles across all organisms and viruses (*Brown et al., 2011*). Protein P30 of the lipid-containing bacteriophage PRD1, structurally determined more than 20 years ago, and predicted as a disordered protein, was the first to be identified possessing a string-like structure running through adjacent virus facets, interacting with capsomers along the edges and cementing them together (*Abrescia et al., 2004*). Giant viruses, however, are structurally more complex, with a larger number of minor capsid proteins holding the virion together beneath the pentasymmetron and trisymmetron. While our current study advances our understanding of this complexity by identifying various ORFs and positioning their models in the density map, deciphering their atomic interactions and specific roles in particle assembly will require higher resolution for the different components.

Jyvaskylavirus isolation is the first step in better understanding the giant virus diversity in the boreal environment. Our preliminary bioprospection attempt using a locally isolated amoebal host suggested a high diversity of unique viral-like particles, while our screening of fish tank samples revealed mixed

populations of other novel viruses with unique morphologies. The origin of the viruses found in aquarium water is unknown, and they could be originated from the groundwater used to keep the fish or the fish microbiome itself. Marseillevirus DNA polymerase sequences are present in metagenomes from Finnish drinking water distribution systems (*Tiwari et al., 2021*) hinting to a wide distribution of these viruses and still unknown ecological role in Central and Eastern Finland. In addition, electron microscopy analyses revealed high structural diversity of giant viruses and virus-like particles in soil, suggesting a gap in knowledge in the diversity and ecology of environmental virus-host interactions (*Hylling et al., 2020*; *Fischer et al., 2023*). Despite the technical challenges that prevented us from characterizing all isolated viruses in this study, our findings underscore the importance that broader sampling and new isolation attempts are essential to determine the distribution of these viruses. The use of host species isolated locally might avoid isolation bias toward the reference strains, giving a better idea of viral diversity and allowing the description of unique viral groups.

In summary, marseilleviruses found in diverse locations and environments worldwide show a strictly conserved architecture, using equivalent pseudo-hexameric building blocks and a plethora of ancillary proteins. We suggest that this adaptation relies on the number, fold, and length of disordered regions in the minor capsid proteins, as well as the membrane composition. These factors not only facilitate correct assembly but would also modulate the structural stability of the virus across various environments.

# Materials and methods

## Key resources table

| Reagent type (species) or resource | Designation | Source or reference | Identifiers | Additional information |
|---|---|---|---|---|
| Strain *Acanthamoeba castellanii* | *Acanthamoeba castellanii* | Aix Marseille University, France | AC | Donated by Prof. Bernard La Scola (Aix Marseille University, France) |
| Strain *Acanthamoeba polyphaga* | *Acanthamoeba polyphaga* | Aix Marseille University, France | AP | Donated by Prof. Bernard La Scola (Aix Marseille University, France). |
| Strain *Vermamoeba vermiformis* | *Vermamoeba vermiformis* | Aix Marseille University, France | AP | Donated by Prof. Bernard La Scola (Aix Marseille University, France) |
| Strain Jyvaskylavirus | Jyvaskylavirus | This study | AC85 | Finnish clade B marseillevirus. Genome accession number: PQ284187 |

## Samples used for screening, amoebal host cultivation, and viral isolation

To get an overview on the presence of giant viruses in Finland and their virus-host interactions, water and soil samples were collected from Jyvaskyla (Finland) and used fresh for viral isolation. Previously collected and stored samples were also used, e.g., aquaculture water samples, recirculating aquaculture filter pellets, and frozen water from experimental aquaria (*Runtuvuori-Salmela et al., 2022*; *Almeida et al., 2019a*; *Almeida et al., 2019b*). Sample details can be found at *Supplementary file 1*. Water samples were collected directly into Eppendorf tubes. Solid samples like soil and composting material (roughly one-fourth of an Eppendorf tube) were resuspended in 1 ml of PAS buffer (120 mg NaCl, 4 mg MgSO₄·7H₂O, 4 mg CaCl₂·2H₂O, 142 mg Na₂HPO₄, 136 mg KH₂PO₄ in 1 l of water) and strongly vortexed for resuspension (*Thomas et al., 2006*). Each sample had the following antibiotic mix added to avoid fungal and bacterial contamination during the isolation process: penicillin (0.14 mg/ml), gentamycin (50 mg/ml), amphotericin B (0.25 µg/ml), ciprofloxacin (0.004 mg/ml), vancomycin (0.004 mg/ml), and doxycycline (0.02 mg/ml). Indicated concentrations are the final concentration of each antibiotic in solution.

Isolation was made using *A. castellanii*, *A. polyphaga,* and *V. vermiformis* hosts. All three host strains were kindly donated by Dr. Bernard La Scola and grown using PYG media at room temperature (~25°C) (*Jensen et al., 1970*). For the isolation process, cells were mixed with 50 µl of the samples in 96-well plates. Cell density was controlled by resuspending confluent T25 flasks in 1 ml of PYG media and diluting it 1/1000 before adding 200 µl to each well in the plates. The antibiotics mentioned above were also present in PYG media used for preparing the isolation plates. Cells were monitored daily for

5 days for the appearance of CPE. In case no effect appeared, the plates were frozen, thawed, and 25 µl of each well used as samples for a second passage. Three passages of all samples were made in all three hosts. Negative samples were those that no CPE appeared after the third passage.

## Growth curve, chloroform sensitivity, and stability

Growth curves were made by infecting confluent cells in 96-well plates with a low multiplicity of infection (MOI) (1/1000 dilution of the virus stock). For each dilution four wells were prepared. At the indicated time points the samples were frozen and then, after two freeze-thaw cycles, 50 µl of each well were transferred to a new 96-well plate containing confluent *A. castellanii* cells for titration. Chloroform sensitivity was made by exposing a 900 µl viral aliquot to 10% chloroform for 10 min, inside an Eppendorf tube. As control, 900 µl of the same viral stock was mixed with 100 µl of PAS and incubated for 10 min. After incubation both were serially diluted and titrated (four replicates per dilution). Stability was evaluated by diluting a viral stock serially (−1 to −11, 1 ml aliquots) and storing the viral dilutions at room temperature (~25°C), cold room (~8°C), and inside an incubator (37°C). Samples were stored for 109 days and then titrated. The incubation time used is this long because of the SARS CoV2-related lockdowns that happened during this experiment. All titers were calculated by tissue culture infectious dose (TCID)50 (*Reed and Muench, 1938*).

## Sequencing and genomic data

For DNA extraction, a confluent T25 was infected at a low MOI and the supernatant was harvested after the appearance of full CPE. Two aliquots of 2 ml each were subjected to nuclease treatment and capsid precipitation by $ZnCl_2$ (*Santos, 1991*). After proteinase K treatment the samples were mixed with ethanol:guanidine and DNA extraction was finished using the GeneJET Genomic DNA Purification Kit (Thermo Fisher). DNA paired-end libraries (2 × 250 bp) were constructed with 1 ng of the viral genome as input using the Nextera XT DNA kit (Illumina, Inc, San Diego, CA, USA) and sequenced on the Illumina MiSeq for 39 hr, the same strategy employed by *Brahim Belhaouari et al., 2022*.

The assembled genome was submitted to ORFs prediction using GeneMarkS (*Besemer et al., 2001*). Also, a search for tRNA genes was performed using ARAGORN program (*Laslett and Canback, 2004*). Only ORFs having more than 150 nucleotides were considered to analysis. Similar sequences for each predicted protein were searched using BLASTp (expect threshold: 10-3) against NCBI nonredundant protein sequences (nr) database. To perform synteny analysis, complete genome sequences from different marseilleviruses were obtained in GenBank. The sequences underwent manual curation to correct distortions caused by the circular topology of marseilleviruses genomes. Then, synteny analysis was conducted using MAUVE program with default parameters (*Darling et al., 2004*).

Phylogenetic trees were constructed based on DNA polymerase and MCP amino acid sequences. The sequences used for alignments were obtained using BLASTp (expect threshold: 10-3) against NCBI nonredundant protein sequences (nr) database. The alignments were performed using MUSCLE executed through MEGA X program (*Edgar, 2004*; *Kumar et al., 2018*). Maximum-likelihood phylogenetic trees were constructed using IQtree software (version 1.6.12) with 1000 bootstrap replicates as branches support (*Kumar, 2018*). Best-fit substitution models were selected by ModelFinder algorithm implemented in IQtree (*Kalyaanamoorthy et al., 2017*). The phylogenetic trees were visualized using iTOL (*Letunic and Bork, 2021*).

## Microscopy

Samples with noticeable CPE during the viral isolation process were prepared for checking the presence of negatively stained viral-like particles. Two microliters of the lysates (supernatant straight from the isolation plate) were added to a microscopy grid and incubated for 2 min at room temperature. The excess liquid was removed with a water-soaked Whatman paper. Then, 5 µl of 2% phosphotungstic acid was added to the grid and its excess was removed after an incubation of 2 min. The grids were left for drying at room temperature for 5 min and used for imaging at a Jeol JEM-1400 electron microscope straight away.

Samples destined for thin sectioning were prepared by infecting a T75 flask with a low MOI. Twenty-four hours later, when the CPE started to appear, cells were harvested from the flask and pelleted by centrifugation (10 min at 2000×*g*). The pellet was resuspended in 2.5% glutaraldehyde 0.1 M sodium phosphate buffer and kept under slow rotation for 1 hr at room temperature. After the incubation

the cells were pelleted again, resuspended in 0.1 M sodium phosphate buffer and sent for blocking and thin section preparation. The samples were also imaged at a Jeol JEM-1400 electron microscope.

Substrates for helium ion microscopy were prepared by incubating silicon chips in poly-l-lysine (Sigma-Aldrich) for 5 min followed by two wash steps in sterile water and left to dry overnight. *A. castellanii* cells were seeded in 24-well plates containing the poly-L-lysine-coated substrates and infected at different MOI. After the appearance of CPE, the samples were fixed with 2.5% glutaraldehyde in 0.1 M sodium cacodylate for 2 hr followed by three washes with 0.1 M sodium cacodylate buffer. After that, the samples were stained by incubation for 1 hr with 1% osmium tetroxide followed by three washes with 0.1 M sodium cacodylate and a second staining with 0.1% tannic acid for 20 min. Dehydration was done with sequential exposure to increasing concentrations of ethanol, using the following percentages: 35%, 50%, 70%, 85%, 95%, and 99%. Each exposure was made for 30 min, except for 99%, which had one 30 min exposure followed by a second overnight exposure to ensure proper dehydration. Ethanol washing was followed up by a critical point drying step using 24 cycles in a Leica EM CPD300 equipment. Macroscopic sample structure did not change during the drying. Imaging was made using the Zeiss Orion Nanofab Helium Ion Microscope from the University of Jyväskylä Nanoscience Center with acceleration voltage 30 kV and ion current 0.2–0.3 pA. Electron flood gun was used during imaging to mitigate positive charging by the ions.

## Virus production for cryo-EM and data collection

A large stock of Jyvaskylavirus was prepared and purified for cryo-EM analysis. Twelve confluent T75 flasks were infected at a low MOI. After the appearance of full CPE the flasks were frozen-thaw to lyse still intact cells, and then all the flask contents were moved to falcon tubes. After one additional freeze-thaw cycle, a brief centrifugation ($500 \times g$, 5 min, 10°C) was made to clear the lysate from cell debris. Then, the viruses were pelleted ($10,000 \times g$, 60 min, 10°C) and resuspended in 700 µl of PAS. The resuspended viruses were loaded into a 10–50% sucrose gradient and centrifuged ($6500 \times g$, 90 min, 10°C). The viral band of the gradient was collected, mixed with PAS buffer to dilute the sucrose, and the viruses were pelleted again. The final pellet was resuspended in 150 µl of PAS buffer, aliquoted and shipped at 10°C to the CIC bioGUNE for cryo-EM analysis. Upon arrival the sample was vitrified using either a Vitrorobot Mark IV (Thermo Fisher) or an Automatic Plunge Freezer EM GP2 (Leica). As a 'quality control' step, some of the grids were inspected using the in-house JEM-2200FS (JEOL, Ltd.) cryo-TEM equipped with a K2 bioquantum camera. The remaining grids were shipped for high-resolution imaging at eBIC - Diamond Light Source (Didcot, UK) in line with democratic access to large infrastructure (*Stuart et al., 2016*). Four data collections on a Titan Krios 300 kV with a K3 camera were performed at a nominal magnification of ×64,000, resulting in a final pixel size of 1.34 or 1.35 Å/pix depending on the microscope used (*Appendix 1—table 1*). Briefly, samples were vitrified on Quantifoil Cu R2/2 or R2/1 300 mesh grids and then collected over 40–45 fractions (with a dose per frame of ≈1 e⁻/Å², with defocus ranges from –0.6 to 3 µm. As we had approximately one particle per hole, different software were used at each data collection to test which strategy would yield the highest number of particles (EPU, TOMO5, and Serial EM; for details in each data collection, see *Appendix 1—table 1*). After four data collections we obtained 3742 useful particles from 17,720 movies.

## 3D reconstruction of Jyvaskylavirus

MotionCorr2 was used to correct the induced beam-shift across the frames within the movies, while the CTF of the individual movies was estimated using the CTFFIND4 software (*Rohou and Grigorieff, 2015*; *Zheng et al., 2017*). Particle auto-picking for the first data collection was performed in crYOLO, training the model with 10 movies *Wagner et al., 2019*; however, the picking process was supervised. For the remaining acquisitions, particles were manually selected in RELION 3.1 due to the limited number of movies available (*Scheres, 2012*). Virions were extracted into a 1000×1000 pixel box, resulting in a final pixel size of 3.087 Å. For each dataset, several rounds of 2D and 3D classifications were performed before 3D refinement, with the initial 3D reference being generated *ab initio* from a limited number of particles from the first data collection. Subsequently, the particles were re-extracted and re-centered, then combined with those from the other data collection after undergoing the same preprocessing workflow. Further classifications led to a homogeneous class comprising 3742 particles (*Appendix 1—table 1*), which then underwent 3D icosahedral refinement with the original

pixel size (1.34 Å/pix) on a 2304 pix box. This was achieved by using the Picasso HPC at the University of Malaga node of the Spanish Supercomputing Network (RES). Computing resources used for the above refinement included 3 tasks, 32 cpu per task, and 1.6 TB of memory (using `--pad` 1). Finally, Ewald sphere correction was applied to the final map, resulting in a resolution of 6.3 Å and a notable improvement in the map interpretability and FSC curve (*Appendix 1—figure 4*).

## Structural analysis of Jyvaskylavirus

Using AlphaFold3 we generated an initial atomic model of the MCP corresponding to the ORF184 identified in this study by sequence comparison with other Marseillaviridae (*Abramson et al., 2024*). The best-ranked predicted model had a confidence level of 95% based on the predicted local-distance difference test, which ranged from 28.7% to 98.9% across the residues. The MCP model was manually fitted into the density corresponding to the pseudo-hexameric capsomer using COOT graphic software (*Emsley and Cowtan, 2004*). The three molecules were rigid body refined into the 6.3 Å resolution map using PHENIX real-space refinement (*Afonine et al., 2018*) leading to a CC$_{mask}$ of 52%.

To identify the penton protein located at the vertices as well as some of the additional visible densities corresponding to the scaffolding and cementing proteins, we used the following strategy. Owing that Jyvaskylavirus belongs to the clade B of the Marseilleviridae family (current study) and that block-based derived cryo-EM maps at the two-, three- and fivefolds of Melbournevirus (clade A) at about 3.5 Å resolution have been recently made available in the Electron Microscopy Database (EMD-37188, -37189, -37190), we used the latest version of ModelAngelo software (v1.0.12) without any fasta sequence input to build and identify the likely models for the penton and ancillary proteins in the Melbournevirus densities (*Burton-Smith and Murata, 2023*; *Jamali et al., 2024*). The sequences of the resulting identified amino acids for the built fragments were then parsed using the HMM routine in ModelAngelo against the list of ORFs of Jyvaskylavirus produced in this study (*Appendix 1—figure 5*). This was performed with the expectation of high sequence homology between the two viruses, as indicated by our comparative sequence analysis (*Figure 2*). For the penton protein, ORF142 was identified, and similarly ORF36, 97, 153, and 121 for the ancillary under the fivefold and cap protein, respectively (*Appendix 1—figure 10*). Then, the corresponding full 3D models were predicted using AlphaFold3 and fitted into the Melbournevirus and Jyvaskylavirus cryo-EM density using the fit-into-map routine in ChimeraX together with the peripentonal capsomers (*Meng et al., 2023*). To assess the metric of this fitting (*Appendix 1—figure 7*), the 3.5 Å fivefold Melbournevirus block 3D density (EMDB-37190) was boxed around the pentameric assembly model and refined as a whole using rigid-body refinement in PHENIX, yielding a CC$_{mask}$ of 57.3%. The same pentameric model was subsequently fitted into the 6.3 Å Jyvaskylavirus 3D cryo-EM density (previously boxed around the model), resulting in a lower CC$_{mask}$ of 33%, consistent with the limited resolution of the capsid map and below regions. In the case of Melbourne virus, the corresponding ORFs were determined by using the Jyvaskylavirus ORFs as templates in BLASTp (https://blast.ncbi.nlm.nih.gov/Blast.cgi?PAGE=Proteins). The predicted models mentioned above were deposited in BioStudies under S-BSST1654.

## Acknowledgements

We would like to thank Mr. Petri Papponen (University of Jyvaskyla, Finland) for help with the electron microscopy, Master of Science. Marjaana Hassani (University of Jyvaskyla, Finland) for donating the composting sample used for Jyvaskylavirus isolation and the staff at the EM Platform at the CIC bioGUNE for infrastructural support and preliminary cryo-grid screening. Prof. Bernard La Scola (Aix Marseille University, France) is thanked for donating the amoeba strains used and for his technical and intellectual input in this project. Yun Song at the UK's national Electron Bio-imaging Centre (eBIC) at Diamond Light Source (DLS) is acknowledged for her assistance in setting up the SerialEM script. We are also grateful to DLS for access and support of the cryo-EM facilities at eBIC (under proposal CM30316), funded by the Wellcome Trust, MRC and BBRSC and to the Spanish Supercomputing Network (RES, Red Española de Supercomputación) for Picasso resources provided by the Supercomputing and Bioinnovation Center (Malaga University) to BCV-2022-3-0015. This study was also funded by the Spanish Ministry of Science, Innovation and Universities via the Severo Ochoa Excellence Accreditation awarded to the CIC bioGUNE (CEX2021-001136-S). JSA is a CNPq researcher. The publication charges for this article have been funded by a grant from the publication fund of UiT The Arctic University of Norway.

## Additional information

### Funding

| Funder | Grant reference number | Author |
|---|---|---|
| Suomen Akatemia | 346772 | Lotta-Riina Sundberg |
| Spanish Ministry of Science and Innovation | PID2021-126130OB-I00 | Nicola GA Abrescia |
| Conselho Nacional de Desenvolvimento Científico e Tecnológico | | Jonatas S Abrahão |
| Coordenação de Aperfeiçoamento de Pessoal de Nível Superior | | Jonatas S Abrahão |
| Fundação de Amparo e Desenvolvimento da Pesquisa | | Jonatas S Abrahão |
| Tromsø Forskningsstiftelse | ID:18_CANS_AS 311192/ A65276 | Gabriel Magno de Freitas Almeida |

The funders had no role in study design, data collection and interpretation, or the decision to submit the work for publication.

### Author contributions

Gabriel Magno de Freitas Almeida, Conceptualization, Data curation, Formal analysis, Investigation, Methodology, Writing – original draft, Writing – review and editing; Iker Arriaga, Data curation, Formal analysis, Investigation, Methodology, Writing – review and editing; Bruna Luiza de Azevedo, Data curation, Formal analysis, Investigation, Writing – original draft, Writing – review and editing; Miika Leppänen, Formal analysis, Investigation, Methodology, Writing – review and editing; Jonatas S Abrahão, Conceptualization, Formal analysis, Investigation, Writing – original draft, Writing – review and editing; Julien Andreani, Data curation, Formal analysis, Writing – review and editing; Davide Zabeo, Formal analysis, Investigation, Visualization, Writing – review and editing; Janne J Ravantti, Data curation, Formal analysis, Investigation, Visualization, Writing – original draft, Writing – review and editing; Nicola GA Abrescia, Conceptualization, Data curation, Funding acquisition, Validation, Investigation, Visualization, Writing – original draft, Writing – review and editing; Lotta-Riina Sundberg, Conceptualization, Resources, Formal analysis, Funding acquisition, Validation, Writing – original draft, Writing – review and editing

### Author ORCIDs

Gabriel Magno de Freitas Almeida ⓘ https://orcid.org/0000-0003-2317-5092
Jonatas S Abrahão ⓘ https://orcid.org/0000-0001-9420-1791
Davide Zabeo ⓘ https://orcid.org/0000-0002-5912-4601
Nicola GA Abrescia ⓘ https://orcid.org/0000-0001-5559-1918
Lotta-Riina Sundberg ⓘ https://orcid.org/0000-0003-3510-4398

Reviewer #1 (Public review): https://doi.org/10.7554/eLife.103492.3.sa1
Reviewer #2 (Public review): https://doi.org/10.7554/eLife.103492.3.sa2
Author response https://doi.org/10.7554/eLife.103492.3.sa3

## Additional files

### Supplementary files

Supplementary file 1. Details of the samples used and isolation results.

Supplementary file 2. Jyvaskylavirus annotation.

MDAR checklist

## Data availability

The Jyvaskylavirus genome has been deposited in GenBank under the accession number PQ284187. Raw stacks movies are uploaded to EMPIAR (EMPIAR-12466), linked to the EMDB accession EMD-51613 corresponding to the reconstructed Jyvaskylavirus cryo-EM map at 6.3 Å resolution. Predicted Jyvaskylavirus PDB models using ModelAngelo and Alphafold have been deposited at BioStudies (https://www.ebi.ac.uk/biostudies/) under the accession number S-BSST1654.

The following datasets were generated:

| Author(s) | Year | Dataset title | Dataset URL | Database and Identifier |
|---|---|---|---|---|
| Almeida GMF, Leppanen M, de Azevedo BL, Abrahao JS, Andreani J, Zabeo D, Arriaga I, Ravantti J, Abrescia NGA, Sundberg L-R | 2025 | Jyvaskylavirus sp. isolate fiAc85, complete genome | https://www.ncbi.nlm.nih.gov/nuccore/PQ284187.1/ | NCBI GenBank, PQ284187 |
| Arriaga I, de Freitas G, Abrescia NGA | 2025 | Raw collected cryo-EM movies | https://www.ebi.ac.uk/empiar/EMPIAR-12466/ | Electron Microscopy Public Image Archive, EMPIAR-12466 |
| Arriaga I, de Freitas G, Abrescia NGA | 2025 | Cryo-EM density map | https://www.ebi.ac.uk/emdb/EMD-51613 | Electron Microscopy Data Bank, EMD-51613 |
| Arriaga I, de Freitas G, Abrescia NGA | 2025 | Predicted PDB models of viral components | https://www.ebi.ac.uk/biostudies/S-BSST1654 | EMBL-EBI BioStudies, S-BSST1654 |

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

# Appendix 1

## Isolation of Finnish giant viruses using a locally isolated amoeba host

During the summer 2019 we isolated a native amoeba from Finland. Amoeba isolation attempts were made by adding a drop of water from different sources to a plate of non-nutrient (NN) agar covered with 100 µl of dead *Escherichia coli* cells. The plates were checked daily for the presence of amoebal-like growth. When cells were seen, a block of agar was cut and added upside down to a fresh NN agar plate containing dead *E. coli*. Plates were then checked daily for the visualization of amoebal cells spreading on the agar from below the agar block. Images of the amoeba isolation process can be seen in *Appendix 1—figure 1A-C*. The NN agar recipe used was 120 mg NaCl, 4 mg MgSO$_4$ · 7H$_2$O, 4 mg CaCl$_2$ · 2H$_2$O, 142 mg Na$_2$HPO$_4$, 136 mg KH$_2$PO$_4$, and 15 g agar in 1 l of water (*Thomas et al., 2006*). Dead *E. coli* cells were prepared by autoclavation of a turbid overnight culture.

For isolating the amoeba 10 freshwater samples were tested. These samples consisted of Jyvasjarvi lake water, water from fish farms, aquarium water, and outlet water from the University of Jyväskylä aquarium facility. All samples had amoeba-like cells growing inside the initial drop. Four samples had amoebal-like cells spreading on the plate from the agar block cut. Of these, one was fast-growing while the other three were growing slow in the following passages. We chose the fast-growing isolate, originated from the University of Jyväskylä aquarium facility, and used it to screen for giant viruses in samples collected from Jyväskylä. We later discovered that this amoebal isolate carried a symbiotic bacterium, making it impossible to grow the amoeba in liquid media. Any attempt resulted in bacterial contamination of the media. Passing the amoeba in antibiotic containing media ended up killing the bacteria and the amoeba at the same time. Encystment of this species was not observed, and preservation by freezing was not effective, meaning that it became impossible to make stocks for future use.

Therefore, further isolation attempts for giant viruses using this amoeba isolate were made in NN agar plates to avoid the problem with liquid media contamination. Fresh amoeba cultures collected from a plate were mixed with the samples to be tested (1:1) and added as drops to a new NN agar plate covered with dead *E. coli* cells. Appearance of CPE led to the harvesting of the sick cells and evaluation of viral-like particles through electron microscopy. Examples of a control culture and two cultures with CPE can be seen in *Appendix 1—figure 1D–F*. CPE in this system resulted in alterations of cell morphology and of patterns of cell distribution on the plate, differing from the typical cell lysis seen in liquid cultures.

Electron microscopy revealed distinct virus-like particles in these enrichment samples (*Appendix 1—figure 1G–K*). However, we were unable to grow these using co-cultures with the original host on agar plates and these samples did not result in CPE when we inoculated them in the other amoebal species used in this study (*A. castellanii, A. polyphaga,* and *V. vermiformis*). Considering that we were unable to grow these putative viruses and consequently could not further characterize them, we show this data here as preliminary evidence of a high diversity of unknown giant viruses in Finland. Host-range and the use of additional locally isolated amoebas should be considered in further studies.

## Jyvaskylavirus growth, chloroform sensitivity, and stability

Jyvaskylavirus grows fast in *A. castellanii* cells (*Appendix 1—figure 2A*). It causes CPE in *V. vermiformis* in high concentrations only (10$^{10}$ TCID50/ml), and in high to middle concentrations in *A. polyphaga* (10$^{10}$–10$^6$ TCID50/ml). Electron microscopy revealed a peculiar double-capsid structure with an indication of a lipid layer in between, so we decided to test whether chloroform would affect the virion stability. Exposure to 10% chloroform for 10 min reduced the viral titer in around six logarithmic orders, indicating the presence of a lipid moiety in the capsid (*Appendix 1—figure 2B*). A stability test was made by leaving viral aliquots at different temperatures for over 100 days. Viruses were recovered from all temperatures including 37°C, showing that the virus is stable for long periods without its host presence (*Appendix 1—figure 2C*).

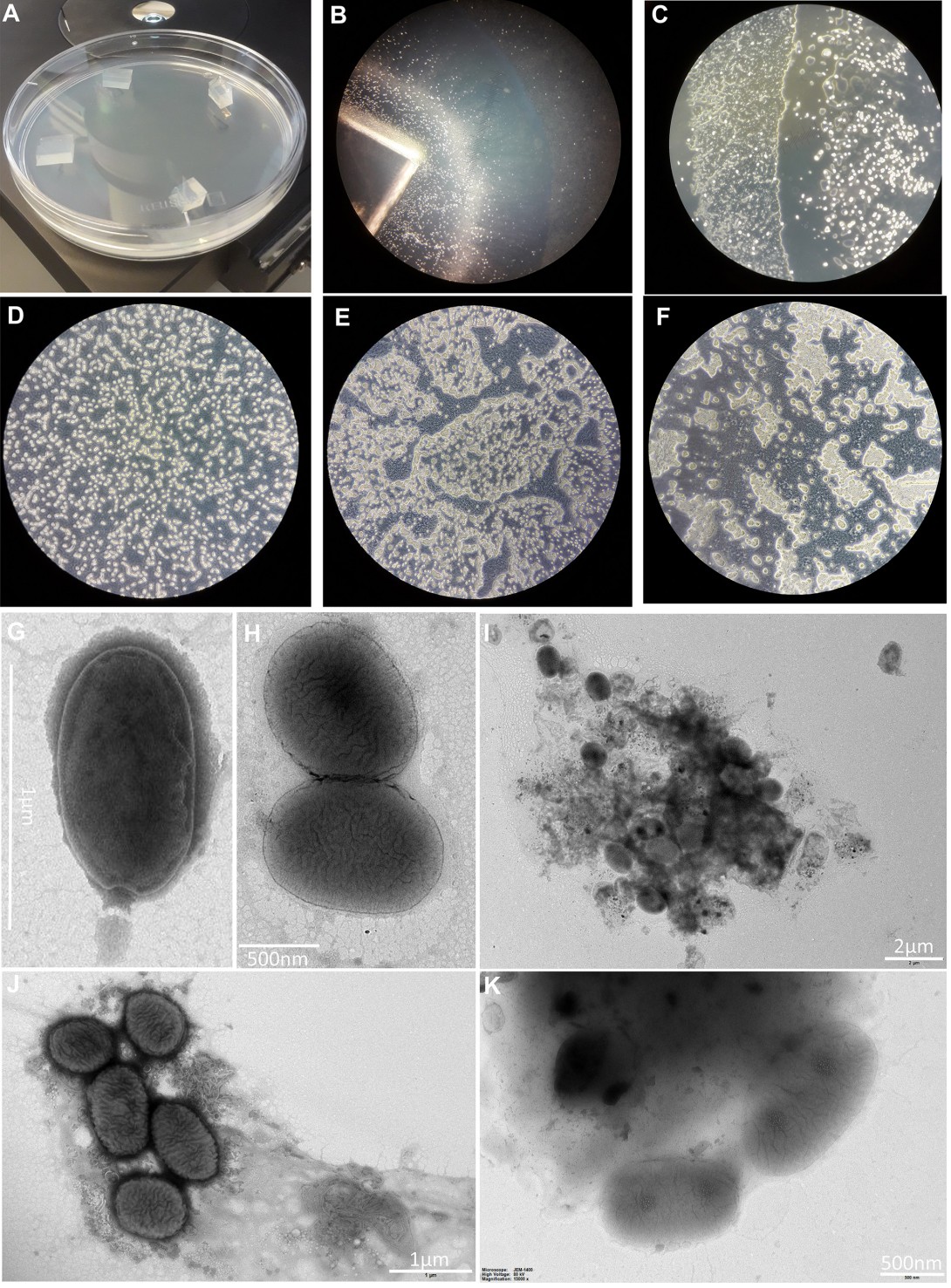

**Appendix 1—figure 1.** Details of our preliminary screening for Finnish giant viruses effort. (**A**) Non-nutrient agar plate used for amoebal isolation. (**B**) Amoebal-like cells spreading from a block of agar. (**C**) Amoebal-like cells growing on top of a non-nutrient agar plate. (**D**) Control culture from an isolation attempt. (**E–F**) Cytopathic effect seen in two different samples. (**G–K**) Viral-like particles seen by electron microscopy.

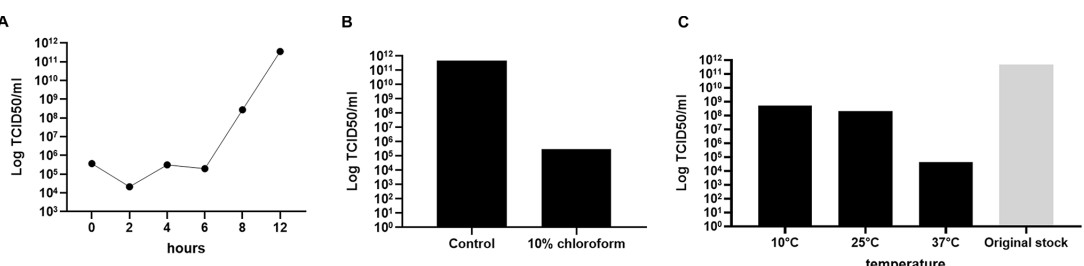

**Appendix 1—figure 2.** Jyvaskylavirus growth curve, sensitivity to chloroform, and stability. (A) Growth curve in *A. castellanii* cells. (B) Jyvaskylavirus sensitivity to chloroform exposure. (C) Jyvaskylavirus stability after storage for 109 days in different temperatures. The original titer at day 0 was $10^{11}$ TCID50/ml. Data shown in A and B were titrated in quadruplicates and in C in octuplicates.

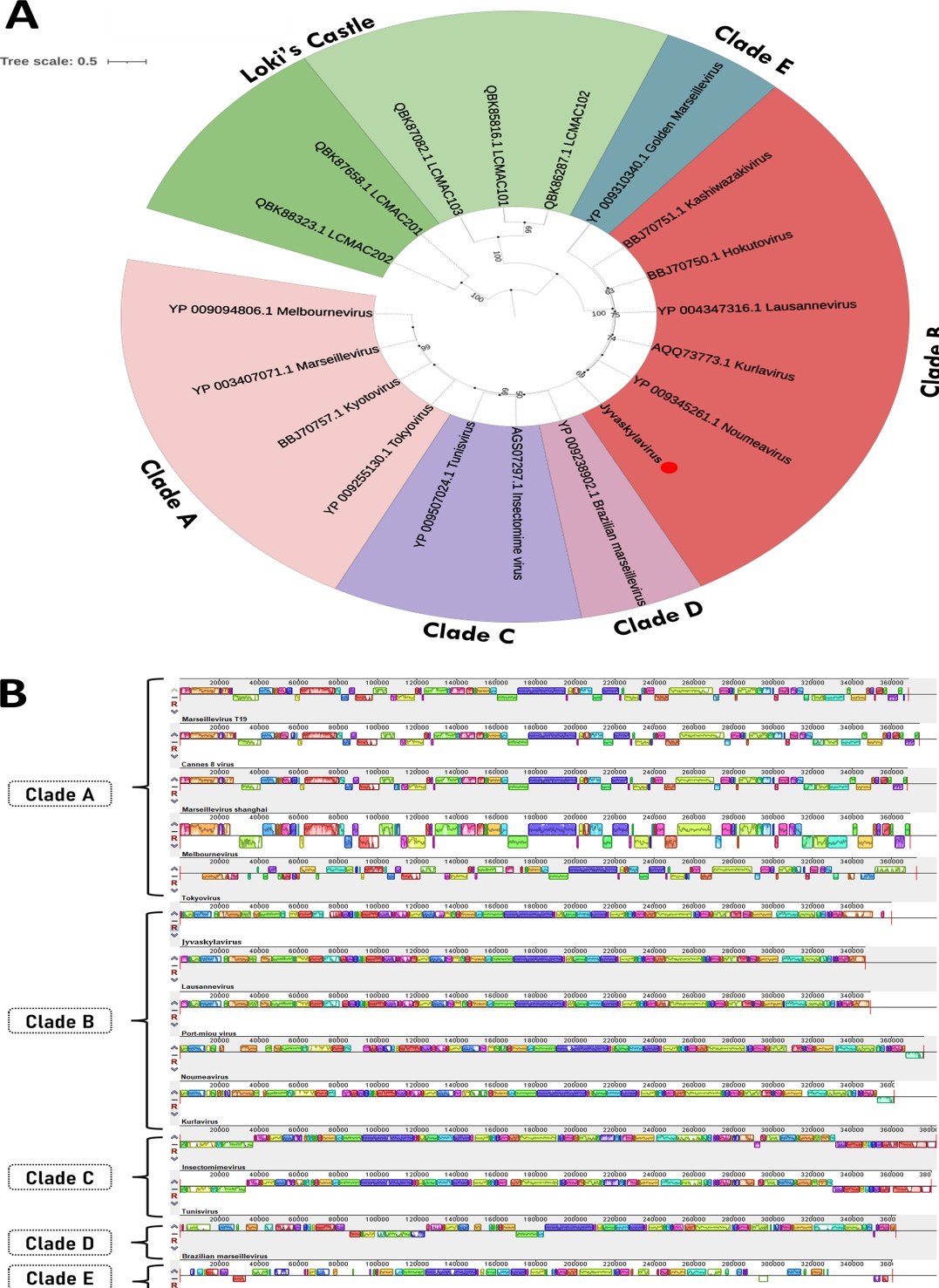

**Appendix 1—figure 3.** Jyvaskylavirus phylogeny and genome synteny. (**A**) Phylogenetic tree reconstructed using major capsid protein (MCP) amino acid sequences belonging to viruses from Marseilleviridae family. The Jyvaskylavirus sequence is highlighted in bold and indicated by a red circle. The alignment was performed with MUSCLE and the maximum-likelihood tree was reconstructed using IQtree software using ultrafast bootstrap (1000 replicates). The best-fit model selected using ModelFinder (implemented in IQtree) was rtREV+F+G4. Scale bar indicates the number of substitutions per site. (**B**) Genome synteny scheme showing the MAUVE alignment of complete genome sequences belonging to marseilleviruses from five different lineages (clades **A–E**). Similar genome blocks are coded by the same color in each sequence. Blocks represented above the x-axis are in the forward strand whereas blocks represented below the x-axis are in the reverse strand.

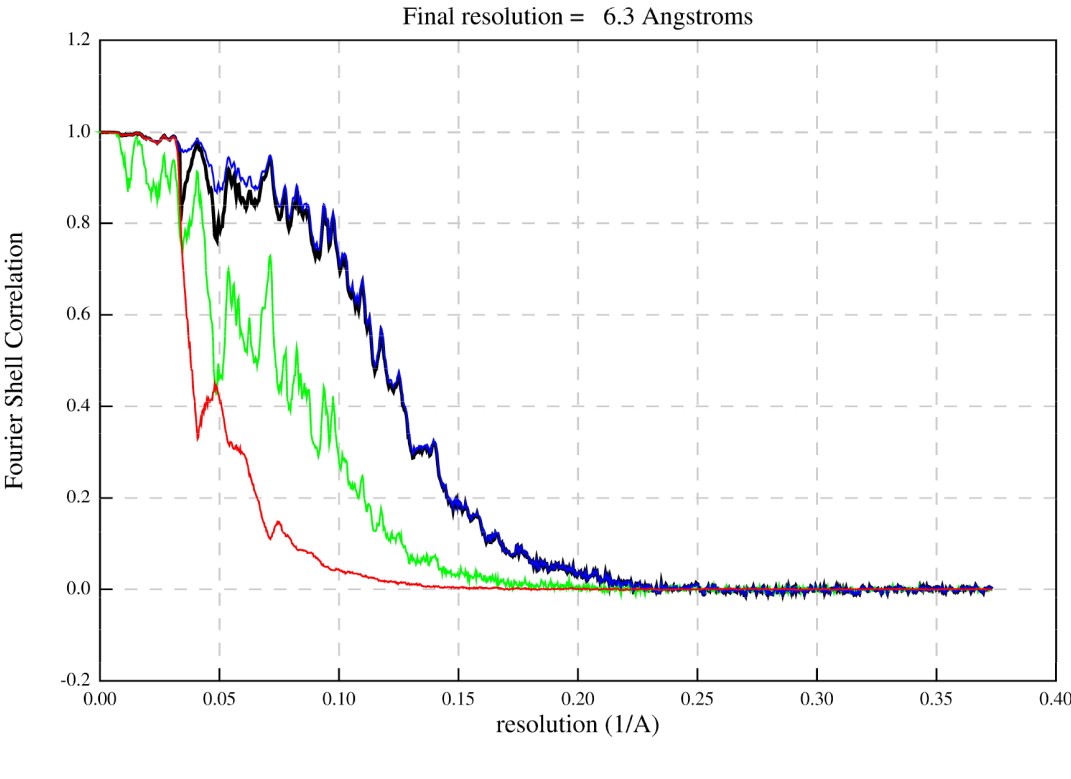

**Appendix 1—figure 4.** Fourier Shell correlation curve of the whole Jyvaskylavirus reporting a resolution of 6.3 Å at the 0.143 criterion (black line: corrected map); other colored curves indicate the following: blue line, masked maps; red line, phase-randomized masked maps; and green line, unmasked maps.

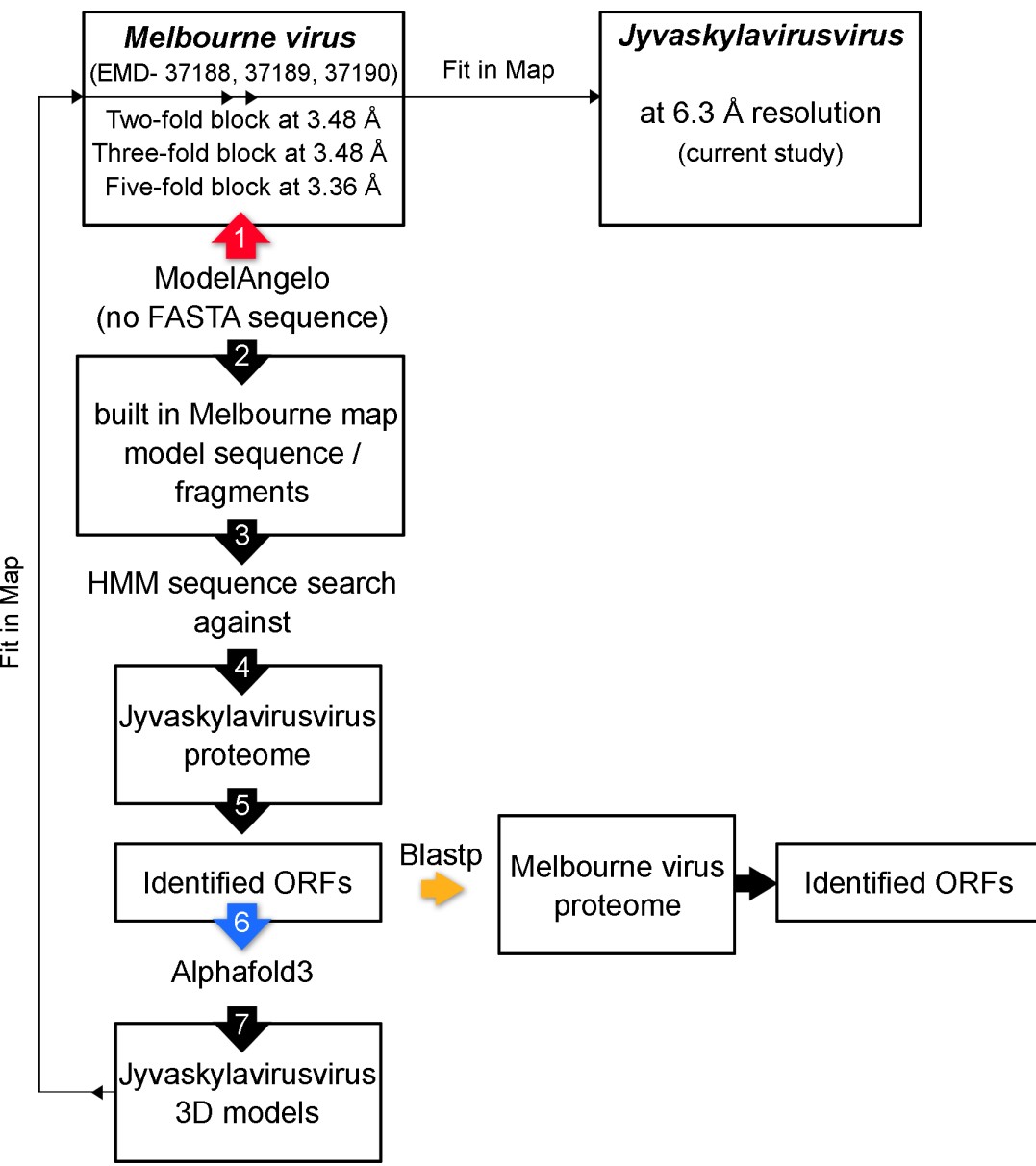

**Appendix 1—figure 5.** Schematic workflow illustrating the identification of additional open reading frames (ORFs) corresponding to ancillary proteins using an integrative approach combining ModelAngelo, AlphaFold3, and ChimeraX software. Large, numbered arrows indicate the sequences of key steps in the workflow: the red arrow marks the use of ModelAngelo to build the polypeptide chains in the Melbournevirus maps; the blue arrow indicates the step of predicting the fold of the identified Jyvaskylavirus ORFs; and the orange arrow marks the step in which the sequences of the Jyvaskylavirus proteins were queried against the reference Melbournevirus isolate 1 (NCBI Reference Sequence: NC_025412.1).

Jyvaskylavirus ORF097 into 5-fold block Melbournevirus map

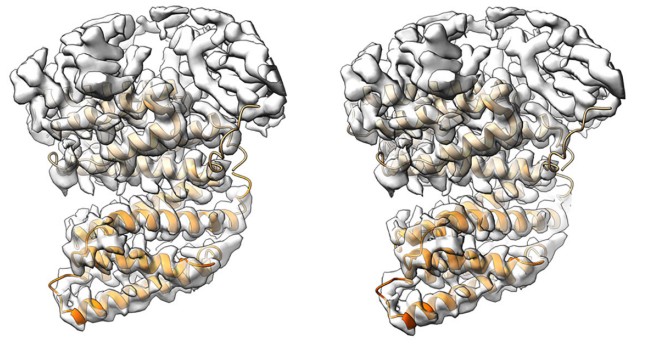

Jyvaskylavirus ORF119 into 5-fold block Melbournevirus map

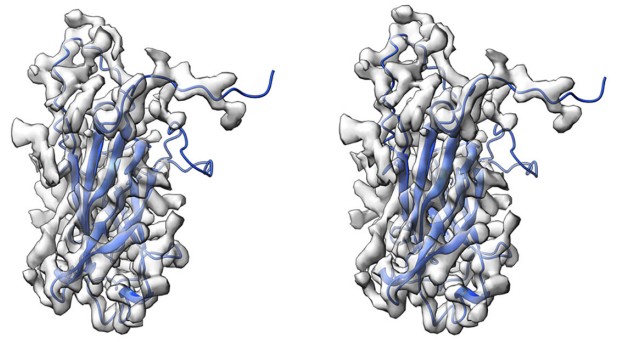

Jyvaskylavirus ORF036 into 5-fold Melbournevirus map

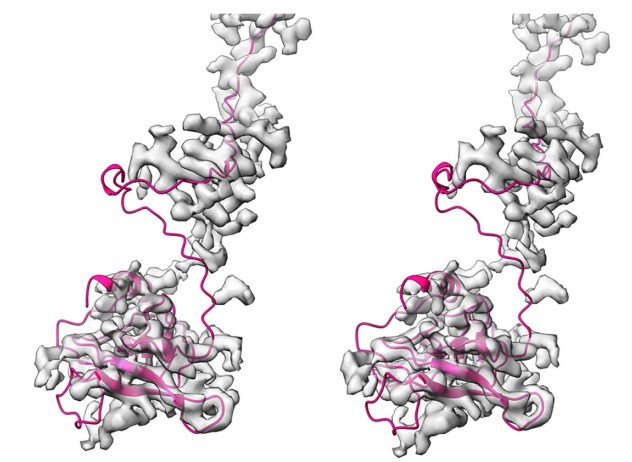

Jyvaskylavirus ORF153 (fragment)
into 5-fold block Melbournevirus map

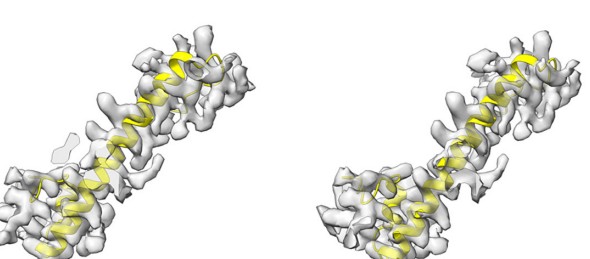

**Appendix 1—figure 6.** Stereo-view (cross-eye) of the identified Jyvaskylavirus open reading frames (ORFs)

*Appendix 1—figure 6 continued on next page*

*Appendix 1—figure 6 continued*

corresponding to the ancillary proteins beneath the capsid shell fitted into the cryo-electron microscopy (cryo-EM) Melbournevirus fivefold block reconstructed density (EMD-37190) shown in semitransparent white; a B-factor of −5 Å$^2$ and resolution cutoff to 4 Å have been applied to the original map for clarity. ORF097 forms a clear dimer while only a helical fragment of ORF153 has been depicted (the inset shows the full prediction). While the predicted core folds of the individual AlphaFold3 models align reasonably well with the density map, the terminal ends, predicted to be flexible, do not fit within the density.

## Jyvaskylavirus ORFs fitted into EMDB-37190 5-fold block Melbournevirus map

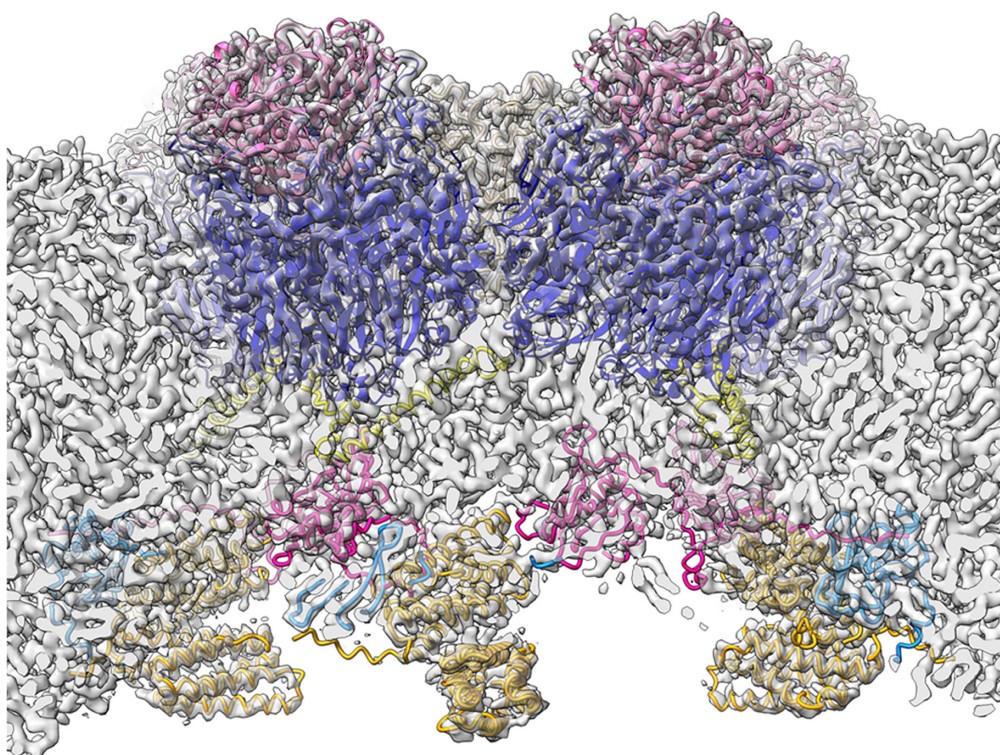

**Appendix 1—figure 7.** Melbournevirus fivefold block reconstructed density (EMD-37190) in semitransparent white (a B-factor of −5 Å$^2$ and resolution cutoff to 4 Å have been applied to the original map for clarity), with the predicted Jyvaskylavirus 3D models of the major capsid protein (ORF184) fitted into the density, composing the peripentonal pseudo-hexameric capsomers represented as spheres in medium blue and capped at the top by ORF121 trimers (shown as pink spheres); five copies of the penton protein (ORF142), represented as cartoon tubes in light-brown color, plug the vertex. Below, different pentasymmetron protein components are represented as cartoon tubes, corresponding to a fragment of ORF153 (yellow), ORF97 (as dimer, color-coded in goldenrod and orange), ORF119 (dodger-blue), and ORF36 (hot pink); not all the density below the vertices is accounted for with the fitted models. Boxing the map around the pentameric assembly atomic model and performing rigid-body fitting yielded a CC$_{mask}$ of 57.3%.

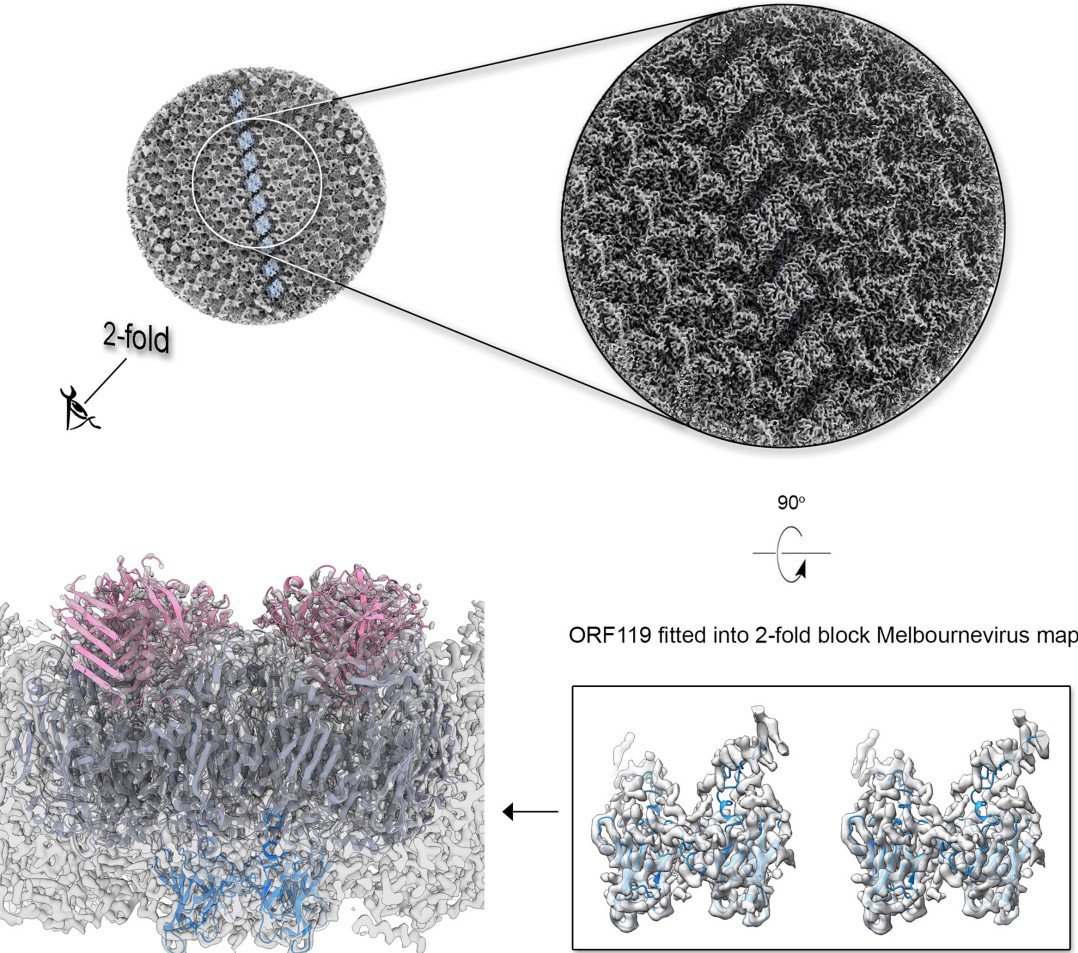

**Appendix 1—figure 8.** Top left, view of the twofold block cryo-electron microscopy (cryo-EM) density reconstruction of Melbourne virus (Gaussian filtered for clarity and rendered as a light gray surface) as seen from within the virion along the twofold icosahedral symmetry axis; densities along the edge of two adjacent trisymmetron are colored in dodger-blue, similar to **Figure 7A** (top left). Top right, enlarged inset corresponding to the region marked by a white circle on the left; this shows the original density with an applied B-factor of –5 Å² and a resolution cutoff of 4 Å for clarity, revealing details of the secondary structure elements of different ancillary proteins. Bottom right, stereoview (cross-eye) of the ORF119 fitted into density showing the matching with the map with the exception of the terminal ends. Bottom left, fitting of two copies of capsomers displayed as in **Appendix 1—figure 7** and viewed perpendicularly to the direction of the twofold axis showing the spatial organization between the copies of ORF119 and the capsomers.

| Predicted ORF142 | PRD1 P31 (1w8x) | HCIV-1 VP9 (6h9c) |
|---|---|---|

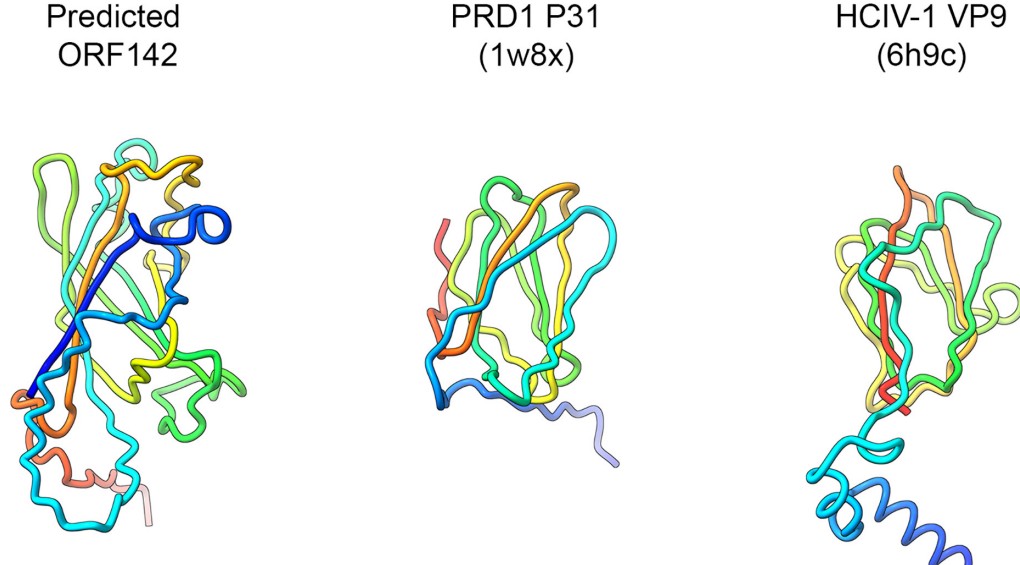

**Appendix 1—figure 9.** A side-by-side comparison of the predicted Jyvaskylavirus ORF142 with experimentally derived penton proteins of the lipid-containing bacteriophage PRD1 and the archaeal virus *Haloarcula californiae* icosahedral virus 1, represented as cartoon tube models color-coded from blue to red in a rainbow gradient from the N-terminal to the C-terminal, with the corresponding PDB ID codes in parentheses.

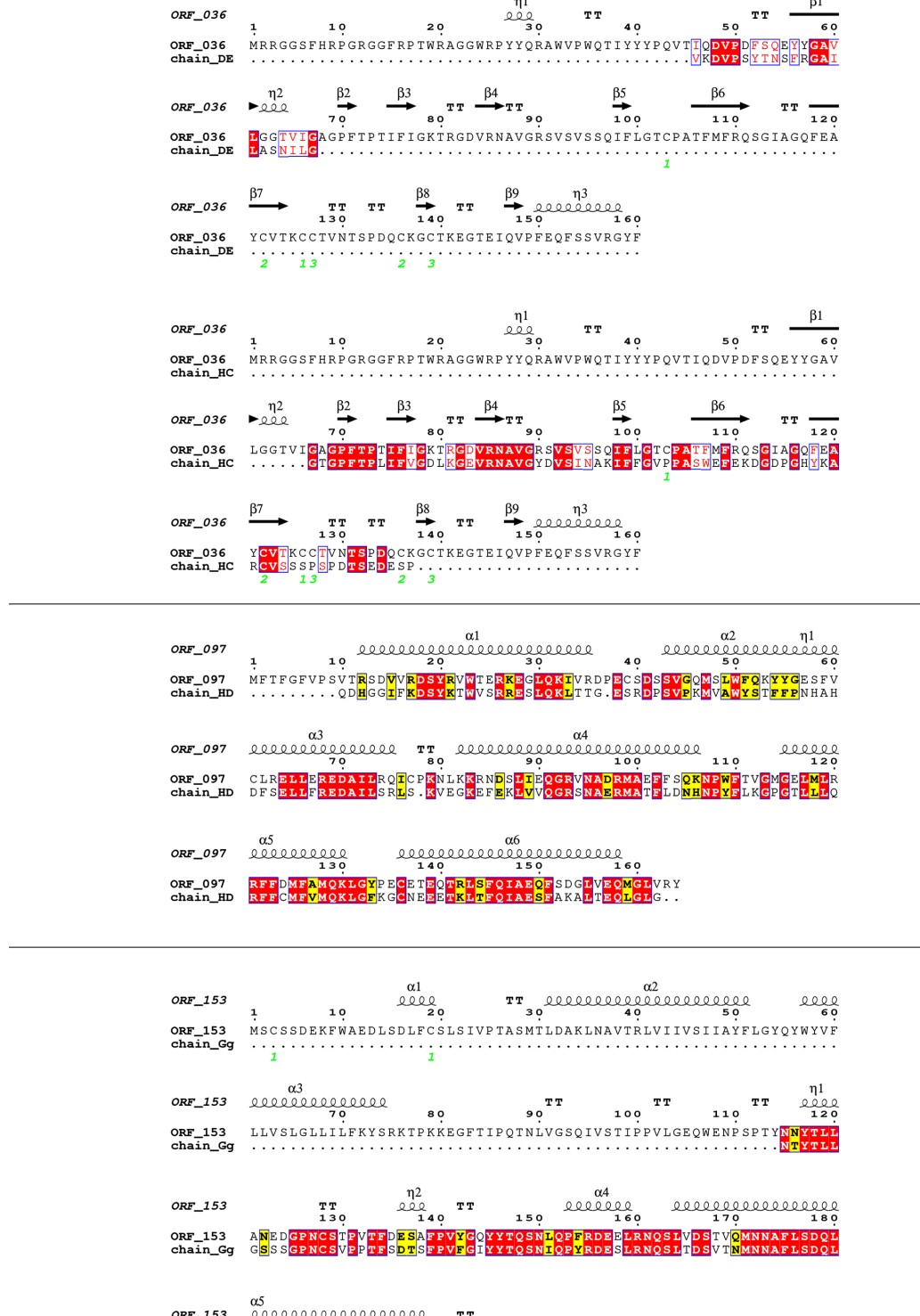

**Appendix 1—figure 10.** Sequence alignment of selected Jyvaskylavirus minor capsid protein sequences (ORF036, ORF097, and ORF153) with residues modeled into the Melbournevirus density map at ~3.5 Å resolution using ModelAngelo, visualized with the ESPRIPT software (https://espript.ibcp.fr/ESPript/ESPript/). Strictly conserved residues are highlighted with red boxes and white characters, while similar residues are marked with yellow boxes and black characters. Secondary structure elements, as predicted by AlphaFold3, are shown above the alignment. Cysteine residues forming disulfide bonds in the predicted structure are indicated by green numbers below the

*Appendix 1—figure 10 continued on next page*

*Appendix 1—figure 10 continued*
alignment. The chains modeled by ModelAngelo are labeled according to the specific fragments that the software could build within the density regions.

**Appendix 1—table 1.** Grid vitrification parameters and cryo-electron microscopy (cryo-EM) data collection.

| Data collection | 1 | 2 | 3 | 4 |
|---|---|---|---|---|
| *Session code @ eBIC* | bi23872-21 | cm30316-1 | cm30316-2 | cm30316-3 |
| *Date* | Jul 2021 | Dec 2021 | Jan 2022 | Mar 2022 |
| *Thermo Fisher Krios 300 kV at eBIC* | Krios IV | Krios II | | |
| *Pixel size at specimen (Å/pix)* | 1.34 | 1.35 | | |
| *SuperRes Bin* | 1 | 2 | | |
| *Collection software* | EPU | Serial EM+ArbitrEM* | TOMO5 2 tilts: [0°, 5°]† | Serial EM |
| *Magnification* | ×64,000 | | | |
| *Detector* | BioQuantum K3 | | | |
| **Data collection and dose parameters** | | | | |
| *No. of frames/fractions* | 40 | 44 | 45 | 43 |
| *Electron fluence (e⁻/Å²)* | 40 | 44.5 | 45 | 45.77 |
| *Electron flux (e²/pix/s)* | 14.7 | 4.8–11.0 (due to ice thickness) | 16.545 | 8 |
| *Exposure time (s)* | 5 | 5.5 | 5 | 10.4 |
| *Defocus range (µm)* | −1.2 to −2.3 | −1 to −2.6 (0.2 step) | −0.6 to −3 (0.3 step) | −1.2 to -2.2 (0.2 step) |
| **Sample preparation** | | | | |
| *Grids* | Quantifoil Cu R2/2 300 mesh | Quantifoil Cu R2/2 300 mesh+Extra carbon layer on top | Quantifoil Cu R2/2 300 mesh | Quantifoil Cu R2/1 300 mesh |
| *Vitrification equipment* | Vitrobot Mark IV (Thermo Fisher) | Automatic Plunge Freezer EM2 (Leica) | Automatic Plunge Freezer EM2 (Leica) | Automatic Plunge Freezer EM2 (Leica) |
| *Vitrification settings* | • Sample droplet: 4 µl<br>• Incubation time: 30 s<br>• Blot time: 2 s<br>• Offset value: −2, −3<br>• Humidity: >95%<br>• Temperature: ≈ 4°C | • Sample droplet: 3 µl<br>• Incubation time: 30 s<br>• Blot time: 0.9 s<br>• Humidity: >90%<br>• Temperature: ≈8°C | • Sample droplet: 3 µl<br>• Incubation time: 30 s<br>• Blot time: 1.5 s<br>• Humidity: >90%<br>• Temperature: ≈8°C | • Sample droplet: 3 µL<br>• Incubation time: 30 s<br>• Blot time: 1.5 s<br>• Humidity: >90%<br>• Temperature: ≈ 8°C |
| **Preprocessing stats** | | | | |
| *No. of movies* | 14,644 | 288 | 383 | 1955 |
| *Extracted particles* | 1608 | 282 | 455 | 2016 |
| *Useful particles* | 1173 | 277 | 437 | 1855 |
| *Particle contributing to the final map* | 3742 | | | |

*ArbitrEM software:https://github.com/kyledent/ArbitrEM (copy archived at **Dent, 2019**).

†The TOMO5 version used for that data collection did not allow to collect a single tilt fraction. Thus, [0°] and [−5°] tilts were collected per view, and then the [−5°] fractions were discarded.

**Appendix 1—table 2.** Identified Melbournevirus open reading frames (ORFs) through Jyvaskylavirus ORFs.

| Jyvaskylavirus | Molecular weight | Melbournevirus | Melbournevirus GeneID | Identity | Role/position |
|---|---|---|---|---|---|
| ORF184 | 52.4 kDa | ORF305 | YP_009094806.1_305 | 96.20% | Major capsid protein (MCP) |
| ORF142 | 22.1 kDa | ORF256 | YP_009094757.1_256 | 83.85% | Penton protein |
| ORF097 | 19.4 kDa | No significant similarity found | Probably incomplete annotation | NA | Pentasymmetron component protein - 1 |
| ORF036 | 17.8 kDa | ORF097 | YP_009094598.1_97 | 73.94% | Pentasymmetron component protein - 2 |
| ORF119 | 14.6 kDa | ORF234 | YP_009094735.1_234 | 85.50% | Pentasymmetron component protein - 3 and trisymmetron facet protein |
| ORF153 | 23.9 kDa | ORF266 | YP_009094767.1_266 | 86.60% | Pentasymmetron component protein - 4 |
| ORF121 | 15.9 kDa | ORF236 | YP_009094737.1_236 | 77.42% | Protein capping the capsomer from above |

**Appendix 1—table 3.** Homologous sequences retrieved by BLASTp when querying Jyvaskylavirus minor capsid protein sequences.

Sequences producing significant alignments in BLASTP:

| Significant alignments in BLASTp for ORF097 | Identity (%) | Accession code | Total score | Query cover | Accession length | E-value |
|---|---|---|---|---|---|---|
| Hypothetical protein LAU 0242 [Lausannevirus] | 99.39 | YP_004347205.1 | 337 | 100% | 164 | 2.00E-11 |
| Hypothetical protein B1750 gp218 [Noumeavirus] | 96.95 | YP_009345363.1 | 330 | 100% | 164 | 9.00E-11 |
| Hypothetical protein A3303 gp283 [Brazilian marseillevirus] | 81.71 | YP_009238788.1 | 286 | 100% | 164 | 3.00E-96 |
| Hypothetical protein ISTM 169 [Insectomime virus] | 80.49 | AHA46067.1 | 278 | 100% | 164 | 6.00E-93 |
| Hypothetical protein D1R32 gp152 [Tunisvirus fontaine2] | 79.27 | YP_009506914.1 | 275 | 100% | 164 | 6.00E-92 |
| Hypothetical protein MarFTME 102 [Marseillevirus futianmevirus] | 61.11 | WRK65147.1 | 195 | 99% | 162 | 2.00E-60 |
| Hypothetical protein [Marseillevirus cajuinensis] | 61.11 | WQM87174.1 | 194 | 99% | 162 | 3.00E-60 |
| Hypothetical protein MAR ORF238 [Marseillevirus marseillevirus] | 61.11 | YP_003406978.1 | 194 | 99% | 160 | 4.00E-60 |
| Hypothetical protein MarSH 247 [Marseillevirus Shanghai 1] | 60.49 | AVR52952.1 | 194 | 99% | 160 | 6.00E-60 |
| Hypothetical protein A9K97 gp203 [Tokyovirus A1] | 59.26 | YP_009255019.1 | 191 | 99% | 160 | 6.00E-59 |
| Hypothetical protein MarFTMF 336 [Marseillevirus sp.] | 57.41 | WNL49852.1 | 189 | 99% | 160 | 6.00E-58 |

| Significant alignments in BLASTp for ORF036 | Identity (%) | Accession code | Total score | Query cover | Accession length | E-value |
|---|---|---|---|---|---|---|
| Hypothetical protein LAU_0176 [Lausannevirus] | 100.00 | YP_004347139.1 | 325 | 100% | 160 | 5.00E-112 |
| Hypothetical protein B1750_gp296 [Noumeavirus] | 95.03 | YP_009345441.1 | 308 | 100% | 161 | 6.00E-105 |
| Hypothetical protein MarXWM_456 [Marseillevirus sp. 'xiwanvirus'] | 93.79 | XCW87046.1 | 306 | 100% | 161 | 2.00E-104 |
| Hypothetical protein A3303_gp208 [Brazilian marseillevirus] | 91.25 | YP_009238713.1 | 286 | 100% | 180 | 5.00E-96 |
| Hypothetical protein D1R32_gp077 [Tunisvirus fontaine2] | 91.93 | YP_009506839.1 | 283 | 100% | 170 | 2.00E-95 |

*Appendix 1—table 3 Continued on next page*

*Appendix 1—table 3 Continued*

**Sequences producing significant alignments in BLASTP:**

| | | | | | | |
|---|---|---|---|---|---|---|
| Hypothetical protein MARUNCVIE_00087 [Marseillevirus sp.] | 73.94 | XAG98523.1 | 253 | 100% | 163 | 2.00E-83 |
| Hypothetical protein A9K97_gp329 [Tokyovirus A1] | 73.94 | YP_009254893.1 | 253 | 100% | 163 | 3.00E-83 |
| Hypothetical protein MAR_ORF113 [Marseillevirus marseillevirus] | 73.94 | YP_003406860.1 | 253 | 100% | 163 | 3.00E-83 |
| Hypothetical protein MarFTMF_478 [Marseillevirus sp.] | 69.54 | WNL49994.1 | 246 | 100% | 174 | 2.00E-80 |
| Hypothetical protein GMAR_ORF101 [Golden Marseillevirus] | 81.88 | YP_009310218.1 | 243 | 100% | 156 | 2.00E-79 |
| Hypothetical protein MarFTME_457 [Marseillevirus futianmevirus] | 72.97 | WRK65502.1 | 235 | 89% | 163 | 4.00E-76 |
| Hypothetical protein [Marseillevirus cajuinensis] | 73.47 | WQM87015.1 | 234 | 89% | 163 | 6.00E-76 |
| Hypothetical protein C8_440 [Cannes 8 virus] | 27.78 | AGV01789.1 | 53.9 | 71% | 126 | 2.00E-05 |
| Hypothetical protein MARUNCVIE_00424 [Marseillevirus sp.] | 27.78 | XAG98860.1 | 53.9 | 71% | 126 | 2.00E-05 |
| Hypothetical protein MarFTME_376 [Marseillevirus futianmevirus] | 27.78 | WRK65421.1 | 50.4 | 71% | 126 | 3.00E-04 |
| Hypothetical protein [Marseillevirus cajuinensis] | 27.78 | WQM86935.1 | 50.4 | 71% | 126 | 4.00E-04 |

| Significant alignments in BLASTp for ORF119 | Identity (%) | Accession code | Total score | Query cover | Accession length | E-value |
|---|---|---|---|---|---|---|
| Hypothetical protein LAU_0272 [Lausannevirus] | 100.00 | YP_004347235.1 | 271 | 100% | 130 | 2.00E-91 |
| Hypothetical protein B1750_gp192 [Noumeavirus] | 98.46 | YP_009345337.1 | 269 | 100% | 130 | 1.00E-90 |
| Hypothetical protein MarXWM_349 [Marseillevirus sp. 'xiwanvirus'] | 98.46 | XCW86939.1 | 269 | 100% | 130 | 1.00E-90 |
| Hypothetical protein D1R32_gp183 [Tunisvirus fontaine2] | 95.38 | YP_009506945.1 | 262 | 100% | 130 | 9.00E-88 |
| Hypothetical protein A3303_gp314 [Brazilian marseillevirus] | 94.62 | YP_009238819.1 | 261 | 100% | 130 | 1.00E-87 |
| Hypothetical protein MarDSR_335 [Marseillevirus sp.] | 98.41 | WNL50374.1 | 260 | 97% | 126 | 3.00E-87 |
| Hypothetical protein GMAR_ORF173 [Golden Marseillevirus] | 93.85 | YP_009310290.1 | 260 | 100% | 130 | 5.00E-87 |
| Hypothetical protein MarFTMF_305 [Marseillevirus sp.] | 85.50 | WNL49821.1 | 236 | 100% | 131 | 8.00E-78 |
| Hypothetical protein A9K97_gp174 [Tokyovirus A1] | 85.50 | YP_009255048.1 | 236 | 100% | 131 | 1.00E-77 |
| Hypothetical protein MarSH_274 [Marseillevirus Shanghai 1] | 85.50 | AVR52979.1 | 235 | 100% | 131 | 3.00E-77 |
| Hypothetical protein MEL_234 [Melbournevirus] | 85.50 | **YP_009094735.1**⇐ | 235 | 100% | 131 | 4.00E-77 |
| Hypothetical protein [Marseillevirus cajuinensis] | 84.25 | WQM87202.1 | 226 | 97% | 127 | 8.00E-74 |
| Hypothetical protein MarFTME_132 [Marseillevirus futianmevirus] | 84.68 | WRK65177.1 | 199 | 85% | 112 | 2.00E-63 |

| Significant alignments in BLASTp for ORF153 | Identity (%) | Accession code | Total score | Query cover | Accession length | E-value |
|---|---|---|---|---|---|---|
| Hypothetical protein LAU_0314 [Lausannevirus] | 100.00 | YP_004347277.1 | 429 | 100% | 209 | 1.00E-151 |
| Hypothetical protein PMV_280 [Port-miou virus] | 99.04 | ALH06978.1 | 426 | 100% | 209 | 4.00E-150 |

*Appendix 1—table 3 Continued on next page*

*Appendix 1—table 3 Continued*

**Sequences producing significant alignments in BLASTP:**

| | | | | | | |
|---|---|---|---|---|---|---|
| Transmembrane domain-containing protein [Noumeavirus] | 96.17 | YP_009345299.1 | 415 | 100% | 209 | 1.00E-145 |
| Tape-measure protein [Marseillevirus sp. 'xiwanvirus'] | 95.69 | XCW86896.1 | 414 | 100% | 209 | 2.00E-145 |
| Hypothetical protein D1R32_gp223 [Tunisvirus fontaine2] | 93.78 | YP_009506985.1 | 413 | 100% | 209 | 4.00E-145 |
| Hypothetical protein A3303_gp359 [Brazilian marseillevirus] | 94.26 | YP_009238864.1 | 390 | 100% | 209 | 8.00E-136 |
| Hypothetical protein A9K97_gp139 [Tokyovirus A1] | 87.08 | YP_009255083.1 | 389 | 100% | 209 | 1.00E-135 |
| Transmembrane domain containing protein [Marseillevirus futianmevirus] | 85.65 | WRK65212.1 | 382 | 100% | 209 | 1.00E-132 |
| Transmembrane domain containing protein [Marseillevirus sp.] | 85.65 | WNL49785.1 | 381 | 100% | 209 | 1.00E-132 |
| Hypothetical protein [Marseillevirus cajuinensis] | 85.17 | WQM87237.1 | 379 | 100% | 209 | 1.00E-131 |
| Hypothetical protein GMAR_ORF199 [Golden marseillevirus] | 90.43 | YP_009310316.1 | 375 | 100% | 209 | 5.00E-130 |
| Membrane protein [Cannes 8 virus] | 86.60 | AGV01658.1 | 358 | 100% | 209 | 3.00E-123 |
| Membrane protein [Marseillevirus marseillevirus] | 86.60 | YP_003407031.1 | 358 | 100% | 209 | 4.00E-123 |
| Hypothetical protein MARUNCVIE_00292 [Marseillevirus sp.] | 87.26 | XAG98728.1 | 268 | 75% | 158 | 2.00E-88 |
| Hypothetical protein LCMAC101_00750 [Marseillevirus LCMAC101] | 36.64 | QBK85488.1 | 124 | 98% | 231 | 8.00E-31 |
| Hypothetical protein [bacterium] | 38.73 | MDB4769441.1 | 113 | 96% | 204 | 7.00E-27 |
| Hypothetical protein LCMAC102_00380 [Marseillevirus LCMAC102] | 34.12 | QBK86243.1 | 110 | 98% | 214 | 1.00E-25 |
| TPA: hypothetical protein [Saprospiraceae bacterium] | 34.88 | HMP28140.1 | 109 | 97% | 216 | 2.00E-25 |
| Hypothetical protein LCDPAC01_02460 [Pithovirus LCDPAC01] | 32.99 | QBK84765.1 | 100 | 90% | 198 | 4.00E-22 |
| Hypothetical protein LCMAC103_02610 [Marseillevirus LCMAC103] | 30.09 | QBK86923.1 | 94.7 | 99% | 214 | 1.00E-19 |
| TPA: hypothetical protein [Nitrosarchaeum sp.] | 29.56 | HSA76080.1 | 85.1 | 96% | 200 | 4.00E-16 |
| Hypothetical protein [Pentanymphon antarcticum iridovirus] | 37.11 | WYL19105.1 | 68.2 | 46% | 154 | 4.00E-10 |
| Putative protein 213L [Iridovirus Liz-CrIV] | 32.29 | QEA08267.1 | 54.7 | 44% | 514 | 2.00E-04 |
| Hypothetical protein IIV31_041L [Armadillidium vulgare iridescent virus] | 44.00 | YP_009046655.1 | 53.9 | 24% | 479 | 4.00E-04 |
| Hypothetical protein LCMAC202_02800 [Marseillevirus LCMAC202] | 46.94 | QBK87919.1 | 52.4 | 23% | 256 | 7.00E-04 |
| 213R [Invertebrate iridescent virus Kaz2018] | 39.22 | QNH08623.1 | 52.8 | 24% | 522 | 9.00E-04 |
| Hypothetical protein [Flavobacteriaceae bacterium] | 34.09 | MBF12528.1 | 51.6 | 52% | 213 | 0.001 |
| Hypothetical protein IIV6-T1_212 [Invertebrate iridescent virus 6] | 39.22 | QMS79462.1 | 51.6 | 24% | 530 | 0.002 |
| 213R [Invertebrate iridescent virus 6] | 39.22 | NP_149676.1 | 51.6 | 24% | 522 | 0.002 |
| Hypothetical protein [Candidatus Colwellbacteria bacterium] | 36.67 | MDD4931402.1 | 48.5 | 27% | 564 | 0.023 |
| Hypothetical protein [Alphaproteobacteria bacterium] | 34.09 | NDA89370.1 | 47.8 | 36% | 238 | 0.029 |

*Appendix 1—table 3 Continued on next page*

*Appendix 1—table 3 Continued*

**Sequences producing significant alignments in BLASTP:**

| Significant alignments in BLASTp for ORF121 | Identity (%) | Accession code | Total score | Query cover | Accession length | E-value |
|---|---|---|---|---|---|---|
| Hypothetical protein PMV_246 [Port-miou virus] | 100.00 | ALH06944.1 | 301 | 100% | 153 | 9.00E-103 |
| Hypothetical protein B1750_gp189 [Noumeavirus] | 97.39 | YP_009345334.1 | 298 | 100% | 153 | 1.00E-101 |
| Receptor binding protein [Marseillevirus sp. 'xiwanvirus'] | 96.73 | XCW86936.1 | 296 | 100% | 153 | 7.00E-101 |
| Hypothetical protein A3303_gp317 [Brazilian marseillevirus] | 96.73 | YP_009238822.1 | 296 | 100% | 153 | 1.00E-100 |
| Hypothetical protein MarXW_YHH_175 [Marseillevirus sp. 'xiwanvirus'] | 96.73 | XKR75189.1 | 295 | 100% | 153 | 3.00E-100 |
| Hypothetical protein D1R32_gp186 [Tunisvirus fontaine2] | 93.46 | YP_009506948.1 | 271 | 100% | 153 | 6.00E-91 |
| Hypothetical protein C8_277 [Cannes 8 virus] | 77.42 | AGV01626.1 | 241 | 100% | 166 | 9.00E-79 |
| Hypothetical protein [Marseillevirus cajuinensis] | 76.77 | WQM87205.1 | 241 | 100% | 155 | 1.00E-78 |
| Hypothetical protein MEL_236 [Melbournevirus] | 77.42 | **YP_009094737.1** ⇐ | 240 | 100% | 155 | 2.00E-78 |
| Hypothetical protein MarFTMF_302 [Marseillevirus sp.] | 78.06 | WNL49818.1 | 238 | 100% | 155 | 1.00E-77 |
| Hypothetical protein MarFTME_135 [Marseillevirus futianmevirus] | 76.13 | WRK65180.1 | 238 | 100% | 155 | 1.00E-77 |
| Hypothetical protein A9K97_gp171 [Tokyovirus A1] | 76.77 | YP_009255051.1 | 235 | 100% | 155 | 2.00E-76 |

The blue colored arrows mark the Melbournevirus hits (in bold).

