## [Editor Report · eLife Assessment]

This manuscript describes an **important** study of the giant virus Jyvaskylavirus. The characterisation presented is **compelling**. The work will be of interest to virologists working on giant viruses as well as those working with other members of the PRD1/Adenoviridae lineage.

---

## [Referee Report · Reviewer #1 (Public review)]

This study presents Jyvaskylavirus, a new member of the Marseilleviridae family, infecting Acanthamoeba castellanii. The study provides a detailed and comprehensive genomic and structural analysis of Jyvaskylavirus. The authors identified ORF142 as the capsid penton protein and additional structural proteins that comprise the virion. Using a combination of imaging techniques the authors provide new insights into the giant virus architecture and lifecycle. The study could be improved by providing atomic coordinates and refinement statistics, comparisons with available giant virus structures could be expanded, and the novelty in terms of the first isolated example of a giant virus from Finland could be expounded upon.

The study contributes new structural and genomic diversity to the Marseilleviridae family, hinting at a broader distribution and ecological significance of giant viruses than previously thought.

Comments on revisions: I'm satisfied with the authors' responses to the review, and request no further changes.

---

## [Referee Report · Reviewer #2 (Public review)]

This paper describes the molecular characterisation of a new isolate of the giant virus Jyvaskylavirus, a member of the Marseilleviridae family infecting Acanthamoeba castellanii. The isolate comes from a boreal environment in Finland, showcasing that giant viruses can thrive in this ecological niche. The authors came up with a non-trivial isolation procedure that can be applied to characterise other members of the family and will be beneficial for the virology field. The genome shows typical Marseilleviridae features and phylogenetically belongs to their clade B. The structural characterisation was performed on the level of isolated virion morphology by negative stain EM, virions associated with cells either during the attachment or release by helium microscopy, the visualisation of the virus assembly inside cells using stained thin sections, and lastly on the protein secondary structure level by reconstructing ~6 A icosahedral map of the massive virion using cryoEM. The cryoEM density combined with gene product structure prediction enabled the identification and functional assessment of various virion proteins. The visualisation of ongoing virus assembly inside virus factories brings interesting hypotheses about the process that; however, needs to be verified in the next studies.

Strengths:

The detailed description of the virus isolation protocol is the largest strength of the paper and I believe it can be modified for isolating various viruses infecting small eukaryotes. The cryoEM map allows us to understand how exceptionally large virions of these viruses are stabilised by minor capsid proteins and nicely demonstrates the integration of medium-resolution cryoEM with protein structure prediction in deciphering virion protein function.

Weaknesses:

No mass spectrometry data are presented to supplement and confirm the identity of virion proteins which predicted models were fitted into the cryoEM density.

---

## [Author Response]

The following is the authors’ response to the original reviews.

**eLife Assessment**
This manuscript describes an important study of the giant virus Jyvaskylavirus. The characterisation presented is solid, although, in the current form, it is not clear to what extent these findings change our perception of how giant viruses, especially those isolated from a cold environment, function. The work will be of interest to virologists working on giant viruses as well as those working with other members of the PRD1/Adenoviridae lineage.

Thank you for the revision and positive comments. We decided to submit our revised version of the manuscript with changes made in light of the comments made by the editorial team and the reviewers. We hope that now the manuscript is in a better shape and satisfies all comments received. Major changes made were:

- We changed the author order considering reviewer 2 comments (point 11). Note that no author was added or removed, we just rearranged the order of authorship.

- We included a new supplementary table with the Jyvaskylavirus genome annotation. This is now supplementary table 2.

- We included a supplementary figure 9 to support our changes based on reviewer 2 comments (point 6).

- Figures 2,5,6,7 and the supplementary figure 2 were updated to accommodate our answers to different reviewer comments.

- Three new references were added to support some of our changes.

Below you will find our responses to each specific point raised by the reviewers.

**Public Reviews:**

**Reviewer #1 (Public review):**
This study presents Jyvaskylavirus, a new member of the Marseilleviridae family, infecting Acanthamoeba castellanii. The study provides a detailed and comprehensive genomic and structural analysis of Jyvaskylavirus. The authors identified ORF142 as the capsid penton protein and additional structural proteins that comprise the virion. Using a combination of imaging techniques the authors provide new insights into the giant virus architecture and lifecycle. The study could be improved by providing atomic coordinates and refinement statistics, comparisons with available giant virus structures could be expanded, and the novelty in terms of the first isolated example of a giant virus from Finland could be expounded upon.The study contributes new structural and genomic diversity to the Marseilleviridae family, hinting at a broader distribution and ecological significance of giant viruses than previously thought.

Thank you for your constructive comments. We have addressed each point raised in our rebuttal letter and revised the manuscript accordingly. By following your specific comments, we improved the manuscript regarding atomic coordinates, refinement statistics and novelty of finding a Finnish marseillevirus. Details are provided in the specific answers to your points.

**Reviewer #2 (Public review):**
Summary:This paper describes the molecular characterisation of a new isolate of the giant virus Jyvaskylavirus, a member of the Marseilleviridae family infecting Acanthamoeba castellanii. The isolate comes from a boreal environment in Finland, showcasing that giant viruses can thrive in this ecological niche. The authors came up with a non-trivial isolation procedure that can be applied to characterise other members of the family and will be beneficial for the virology field. The genome shows typical Marseilleviridae features and phylogenetically belongs to their clade B. The structural characterisation was performed on the level of isolated virion morphology by negative stain EM, virions associated with cells either during the attachment or release by helium microscopy, the visualisation of the virus assembly inside cells using stained thin sections, and lastly on the protein secondary structure level by reconstructing ~6 A icosahedral map of the massive virion using cryoEM. The cryoEM density combined with gene product structure prediction enabled the identification and functional assessment of various virion proteins.Strengths:The detailed description of the virus isolation protocol is the largest strength of the paper and this reviewer believes it can be modified for isolating various viruses infecting small eukaryotes. The cryoEM map allows us to understand how exceptionally large virions of these viruses are stabilised by minor capsid proteins and nicely demonstrates the integration of medium-resolution cryoEM with protein structure prediction in deciphering virion protein function. The visualisation of ongoing virus assembly inside virus factories brings interesting hypotheses about the process that; however, needs to be verified in the next studies.Weaknesses:The conclusions from helium microscopy images are overinterpreted, as the native membrane structure cannot be preserved in a fixed and dehydrated sample. In the image, there are many other parts of the curved membrane and a lot of virions, to me it seems the specific position of the highlighted virion could arise by a random chance. The claim that the cells were imaged in the near-original state by this method should be therefore omitted. Also, no mass spectrometry data are presented that would supplement and confirm the identity of virion proteins which predicted models were fitted into the cryoEM density. For a general virology reader outside of the giant virus field, the results presented in the current state might not have enough influence and the section should be rewritten to better showcase the novelty of findings.

Thank you for your constructive comments. We thank reviewer #2 for highlighting these weaknesses, giving us the opportunity to improve our study. We have removed the claim that the cells were imaged in a near-original state. Additionally, we agree that the positions of the virions on the cell surface could result from a random distribution. However, the specific virion in panel 3C is situated halfway into a crevice, and it cannot be ruled out that this particular one could be in the process of being endocytotically uptaken. This is why we used the term "probably" while referring to this finding. Regarding the mass spectrometry data, while we understand that MS data would provide an additional layer of evidence to validate the specific proteins present in the virion, they would not confirm the precise location or role of these proteins within the virion.

We have addressed each point raised in our rebuttal letter and revised the manuscript accordingly.

**Recommendations for the authors:**

**Reviewer #1 (Recommendations for the authors):**
I have only minor comments which should be relatively simple to address:(1) Atomic coordinates should be deposited in the PDB, and refinement statistics for the models provided, for example by expanding Table S2.

We thank reviewer #1 for the suggestion. In the original submission in the ‘Data availability’ statement we stated that ‘Predicted Jyvaskylavirus PDB models using ModelAngelo and Alphafold have been deposited at BioStudies under the accession number S-BSST1654’. So, atomic coordinates of all predicted models are publicly available at the https://www.ebi.ac.uk/biostudies/ ; for additional clarity we also added the link in the ‘Data availability’ statement in the revised version.

Our reasoning of not depositing them in the Protein Data Bank associated to our EMD-51613 entry is because they remain predicted models rigid-body fitted into the Jyvaskylavirus density map of 6.3 Å resolution. However, we have added into our BioStudies deposition (BSST1654) the whole Jyvaskylavirus pentameric assembly model (including all identified and predicted major and minor capsid proteins) rigid-body fitted into the Jyvaskylavirus map, and it can be easily downloaded.

We did not to perform the real-space ‘minimization_global’ refinement of the predicted models corresponding to the ORFs of Melbournevirus (or Jyvaskylavirus) into the corresponding Melbournevirus available densities with entries EMD-37188, 37189, 37190 at ~ 3.5 Å resolution (by block-based reconstruction methods) as these maps were generated and deposited by other authors. Instead, we performed the rigid-body fit-into-map procedure of the individual predicted Jyvaskylavirus models into the previously deposited Melbournevirus maps using ChimeraX, demonstrating a fold-map alignment and assignment (see for example the individual stereo views in Supplementary Figure 6).

In the revised version, we now provide the refinement statistics for the complete Jyvaskylavirus pentameric assembly (inclusive of peripentonal major capsid and minor capsid proteins) rigid-body fitted as a whole into the Melbournevirus 5-block reconstruction map using PHENIX, resulting into a CC_mask_ of 57.3% (this is also stated in Supplementary Figure 7). The same pentameric assembly model was then placed into our lower-resolution 6.3 Å Jyvaskylavirus 3D density map in ChimeraX and rigid-body refined as a whole in PHENIX, yielding a predictably lower CC_mask_ of 33%. This pentameric assembly model has now also been included into BioStudies entry.

The procedure for this rigid body fitting and refinement has been clarified and added to the 'Materials and Methods' section as follows:

“Then, the corresponding full 3D models were predicted using AlphaFold3 and fitted into the Melbournevirus and Jyvaskylavirus cryoEM density using the fit-into-map routine in ChimeraX together with the peripentonal capsomers (Meng et al 2023). To assess the metric of this fitting (Supplementary Figure 7), the 3.5 Å five-fold Melbournevirus block 3D density (EMDB-37190) was boxed around the pentameric assembly model and refined as a whole using rigid-body refinement in PHENIX, yielding a CC_mask_ of 57.3%. The same pentameric model was subsequently fitted into the 6.3 Å Jyvaskylavirus 3D cryo-EM density (previously boxed around the model), resulting in a lower CC_mask_ of 33%, consistent with the limited resolution of the capsid map and below regions.”

(2) The results section 'Jyvaskylavirus three-dimensional architecture' could be expanded to compare and contrast with other giant virus structures, in terms of T-number, diameter, and features on and inside the capsid. This is not essential but would help focus claims of novelty with regard to structure.

We have added a few lines as indicated by reviewer#1 to contextualize in morphological terms Jyvaskylavirus with other NCLDV viruses as follows:

“Both the capsid organization and virion size are similar to those of other Marseilleviruses, such as Melbournevirus and Tokyovirus. Pacmanvirus, considered to be at the crossroads between Asfarviridae and Faustoviruses, also possesses the same T number (309) and a comparable diameter to Jyvaskylavirus. In contrast, other giant viruses, such as African swine fever virus (ASFV), representative of the Asfarviridae family, have a T number of 277 and a diameter of approximately 2,100 Å, while PBCV-1, a member of the Phycodnaviridae family, has a T number of 169 and an average diameter of 1,900 Å. All of the above-mentioned viruses have been shown to possess a major capsid protein with a vertical double jelly-roll fold that composes the capsid shell, along with an internal membrane bilayer. Minor capsid proteins have been identified and structurally modelled for the smaller virions ASFV and PBCV-1 (Wang et al. 2019; Shao et al. 2022).”

(3) The authors highlight one of the main novelties of the virus as being the first to be isolated from Finland. The first isolation of a giant virus from the region is indeed a success but reported isolation experiments for giant viruses are still relatively few. To help shed light on the likely distribution of Jyvaskylavirus-like viruses in the region, and further afield, the genome of Jyvaskylavirus could be searched against relevant available metagenomes.

In the last decade the interest on finding giant viruses by metagenomics has increased. However, the focus has been on marine environments, where these viruses are shown to be prevalent. Besides the few isolates from the Northern hemisphere mentioned in the manuscript, northern giant viruses were detected in metagenome datasets from glacier samples, epishelf lakes, the permafrost, the Nordic seas and in a deep-sea hydrothermal vent. Most of the genomic hits are for mimivirus-like or phycodnavirus-like sequences. A few marseilleviruses were found in the Loki’s castle deep sea vent, and we have already included these sequences in the analysis shown by the supplementary figure 3. In this case the deep-sea vent viruses clusters outside the conventional clades of the marseilleviridae family, evidencing their uniqueness.

In response to the suggestion of exploring the distribution of Jyvaskylavirus, we utilized the MGnify-database to search for DNA polymerase (DNApol) and major capsid protein (MCP) sequences. Our findings revealed multiple hits with significantly low E-values (< 1e-80), where both DNApol and MCP were detected from the same studies, indicating the presence of similar virus-like particles (VLPs) globally. Of particular interest was the detection of similar sequences in metagenomes and transcriptomes obtained from drinking water distribution systems of ground and surface waterworks in central and eastern Finland (https://www.ebi.ac.uk/metagenomics/studies/MGYS00005650#overview). We have acknowledged this in the manuscript and cited the appropriated references, as follows:

Results: “Searching the Jyvaskylavirus major capsid protein and DNA polymerase sequences in the MGnify-database (Richardson et al 2023) yields multiple hits with significantly low E-values (< 1e-80), as expected from the apparent ubiquity of marseilleviruses. Of note was the detection of similar sequences in metagenomes and transcriptomes obtained from drinking water distribution systems of ground and surface waterworks in central and eastern Finland, evidencing that marseilleviruses are prevalent but still unexplored in this region (Tiwari et al 2022)”.

Discussion: “Marseillevirus DNA polymerase sequences are present in metagenomes from Finnish drinking water distribution systems (Tiwari et al 2022), hinting to a wide distribution of these viruses and still unknown ecological role in Central and Eastern Finland.”

**Reviewer #2 (Recommendations for the authors):**
Apart from the major comments in the weaknesses section, I have these additional minor comments to the authors:(1) I do not understand why the authors emphasized the uniqueness of isolating a giant virus from Finland. I think the manuscript would benefit if they rather emphasize that the virus comes from a boreal environment.

The first giant virus, APMV, was described in 2003. In the following years the apparent ubiquity of these viruses was evidenced by two fronts. Metagenomics made clear that giant viruses are found almost everywhere, biased towards the oceans. Isolation efforts brought new virus groups in evidence but has been so far biased towards central Europe and South America samples. The closest isolated giant viruses to Jyvaskylavirus would be either an uncharacterized Swedish cedratvirus or a few microalgae-infecting mimivirus-like and phycodnaviruses-like isolates from Norway. Among marseilleviruses, Jyvaskylavirus is the northernmost isolate so far. Other marseilleviruses from the northern hemisphere were found in France, India, Japan and Algeria only.

We still believe that finding a giant virus in Finland is relevant, considering that no other is known to date, be as an isolate or detected by genomics. We have made these observations clearer in the manuscript, giving emphasis to the boreal environment as well.

(2) All discussed AlphaFold models should be added as Supplementary PDB data.

We thank reviewer #2 for the suggestion. In the original submission in the ‘Data availability’ statement we stated that ‘Predicted Jyvaskylavirus PDB models using ModelAngelo and Alphafold have been deposited at BioStudies under the accession number S-BSST1654’. So, atomic coordinates of all predicted models are publicly available at the https://www.ebi.ac.uk/biostudies/ ; for additional clarity we also added the link in the ‘Data availability’ statement in the revised version.

Our reasoning of not depositing them in the Protein Data Bank associated to our EMD-51613 entry is because they remain predicted models rigid-body fitted into the Jyvaskylavirus density map of 6.3 Å resolution. However, we have added into our BioStudies deposition (BSST1654) the whole Jyvaskylavirus pentameric assembly model (including all identified and predicted major and minor capsid proteins) rigid-body fitted into the Jyvaskylavirus map, and it can be easily downloaded.

We did not to perform the real-space ‘minimization_global’ refinement of the predicted models corresponding to the ORFs of Melbournevirus (or Jyvaskylavirus) into the corresponding Melbournevirus available densities with entries EMD-37188, 37189, 37190 at ~ 3.5 Å resolution (by block-based reconstruction methods) as these maps were generated and deposited by other authors. Instead, we performed the rigid-body fit-into-map procedure of the individual predicted Jyvaskylavirus models into the previously deposited Melbournevirus maps using ChimeraX, demonstrating a fold-map alignment and assignment (see for example the individual stereo views in Supplementary Figure 6).

In the revised version, we now provide the refinement statistics for the complete Jyvaskylavirus pentameric assembly (inclusive of peripentonal major capsid and minor capsid proteins) rigid-body fitted as a whole into the Melbournevirus 5-block reconstruction map using PHENIX, resulting into a CC_mask_ of 57.3% (this is also stated in Supplementary Figure 7).

The same pentameric assembly model was then placed into our lower-resolution 6.3 Å Jyvaskylavirus 3D density map in ChimeraX and rigid-body refined as a whole in PHENIX, yielding a predictably lower CC_mask_ of 33%. This pentameric assembly model has now also been included into BioStudies entry.

The procedure for this rigid body fitting and refinement has been clarified and added to the 'Materials and Methods' section as follows:

“Then, the corresponding full 3D models were predicted using AlphaFold3 and fitted into the Melbournevirus and Jyvaskylavirus cryoEM density using the fit-into-map routine in ChimeraX together with the peripentonal capsomers (Meng et al 2023). To assess the metric of this fitting (Supplementary Figure 7), the 3.5 Å five-fold Melbournevirus block 3D density (EMDB-37190) was boxed around the pentameric assembly model and refined as a whole using rigid-body refinement in PHENIX, yielding a CC_mask_ of 57.3%. The same pentameric model was subsequently fitted into the 6.3 Å Jyvaskylavirus 3D cryo-EM density (previously boxed around the model), resulting in a lower CC_mask_ of 33%, consistent with the limited resolution of the capsid map and below regions.”

(3) Figure 2A: Could ORFs that encode structural proteins discussed in the paper, be somehow highlighted?

We have updated Figure2A to include this information.

(4) Figure 2C: Could be somehow highlighted from these members on which there was conducted structural characterisation (e.g. by some symbol next to the name)?

We have updated Figure2C to include this information.

(5) Figure 5A: Could the central bid be shown in a lower threshold (you can retain the threshold for the protein shell)? It would be interesting to see some details of the interior, rather than a massive blob.

We have decreased the threshold level of the map as suggested.

(6) Figure 6: the density corresponding to MCPs, minor capsid, and penton proteins respectively could be colour-zoned in Chimera(X). This would better visualise where each entity lies.About ORF142 - what other virus protein possesses this fold? Is it similar to the penton protein in other PRD1/Adenoviridae viruses? Maybe some comparison could be presented?

We have incorporated the feedback from reviewer_#_2 by modifying the corresponding panel A in Figure 6. We have colour-zoned the penton (ORF142), some of the density region corresponding to the MCPs (ORF184) and to the minor cap proteins (ORF121). We have kept in grey the density corresponding to other minor proteins, and those we were able to identify are logically introduced later and shown as individual coloured cartoon tube models fitted into the density in panel A of Figure 7.

Regarding ORF142, we have included a reference in the Discussion section to a new Supplementary Figure 9, where we provide a side-by-side comparison of the predicted Jyvaskylavirus penton protein model with experimentally derived penton protein models of PRD1 and HCIV-1. In light of this comparison, we have also added a brief clarification in the Discussion as follows:

“However, in ORF142, the CHEF strands are predicted to be tilted relative to the BIDG strands, with an estimated angle of approximately 60° based on visual inspection (Supplementary Figure 9).”

(7) Figure 7B: Could the density around the protein be zoned (rather than side view clipped), as this would better showcase how it fits the density?

Initially, we presented a side view of the clipped surface to highlight the correspondence between the wall-shaped density, characteristic of a low-resolution beta-barrel, and the beta-barrel of the predicted model. Following the Reviewer’s suggestion, we have now surface-zoned the density and provided a stereo view of the density with the model fitted into the map using ChimeraX. While we recognize that stereo views are no longer commonly used in main text figures, we believe they remain valuable for visually assessing the overall match in low-resolution 3D density maps.

(8) The authors did not try to reconstruct the asymmetric feature of the virion by classifying pentons, which may have identified a special vertex, one they claim might be required for genome packaging in "open particles". I understand the number of particles is low, but even low-resolution classification in C5 might be of interest in the field.

We thank reviewer #2 for this valuable comment. The potential existence of a unique vertex in Marseilleviruses remains an open and intriguing question. Further investigations, including a significant increase in the number of particles, may help clarify this issue, and we plan to explore this topic in future structural studies.

(9) Supplementary Figure 2: It would be interesting how the titre changes after the 12 hours, will it plateau? Could you add a bar showing the original titre to the chart showing stability after 109 days? I like the data in this figure and think it should be transferred to the main text.

The titre at the 12h time point is very close to the titre we often get in our stocks, indicating that indeed it is close to peaking. For comparison: the titre of the 12-hour time point was 10^11.55^ TCID50/ml, whereas our stock has a titre of 10^11.66^ TCID50/ml. Our growth curve had more time points up to 48h and we lost the later time points due to a higher viral load than predicted, which led to us not being able to count these time points with the dilutions used. Showing the first 12 hours was enough for our initial purpose, which was to show a quick replication cycle for Jyvaskylavirus, in accordance with the other marseilleviruses in which the timing of the replication cycle was observed (see the answer for point 10 below).

We have added a bar representing the original titre of the stock used for the stability experiment as suggested.

While preparing the draft we were divided into having the growth and stability figure in the main text or in the supplementary material. Our decision was to move this data to the supplementary material and keep the focus of the main text on the discovery, genome analysis and structural data, as these are the main findings of our work. The specifics regarding stability, growth and other uncharacterized VLPs went to the supplementary material for those in the field who are interested in looking deeper. That being said, we will decide to keep this data as supplementary material if you and the editor agrees.

(10) In the Discussion, the authors should focus on how our perception of giant viruses changes by this study - compare with other growth curves, stability assays, and structures of giant viruses, showcasing how prevalent those stabilising minor capsid proteins are, etc. My impression is that in the current form, it is just not clear if/how substantial these findings are and such a comparison and putting the results in a bigger picture would considerably increase the impact of the paper.

Our comparisons with other marseilleviruses were based on genomic and structural characteristics, the two fronts we had data from the literature and databases to compare to. Sadly there is not too much information regarding stability and growth of other isolates that could be used for an in-depth comparison. For example: although marseilleviruses are known to have a fast replication cycle, this has been measured by DAPI staining of DNA inside infected cells to evaluate viral factory formation (Boyer et al 2009), or by time-series observations of viral cycle stages by electron microscopy (Fabre et al 2017), and not by viral titration as done here. We included a mention to these references in the results:

“A fast replication cycle is a feature also shown for other marseilleviruses (Boyer et al 2009 ; Fabre et al 2017).”

The literature also does not show virion stability of other isolates, making it impossible to have a comparison with jyvaskylavirus. A comparative study testing different isolates side by side is definitely of relevance and interest, but this would be difficult to be done in a short time due to obtaining other isolates. We believe the results in this manuscript might set some parameters to be used for comparing with other marseilleviruses, by our groups and others, in the future.

Regarding the prevalence of the minor capsid proteins, we have expanded and clarified the identification of ORFs in Melbournevirus in the ‘Results’ and ‘Discussion’ sections. The revised Supplementary Table 4 has been updated accordingly and referenced in the results to clarify that the identification of Melbourne ORFs was carried out in BLASTp by querying the Jyvaskylavirus minor protein sequences exclusively against the Melbournevirus isolate 1 (NCBI Reference Sequence: NC_025412.1). BLASTp was then performed against the full sequence database, and homologous sequences were primarily retrieved from other marseillaviruses. These results have been compiled in a new Supplementary Table 5.

However, Supplementary Table 5 also shows that the hits for Melbournevirus are not ranked at the top, and in some cases, they do not appear among the top hits.

The ‘Results’ section now contains the following text:

“To this end, we identified the corresponding Jyvaskylavirus ORFs in Melbournevirus through sequence comparison with Melbournevirus isolate 1 (NCBI Reference Sequence: NC_025412.1) (Supplementary Table 34). However, when the identified Jyvaskylavirus ORF sequences were analyzed using BLASTp without restricting the search to the Melbournevirus reference, many hits were observed in other giant viruses, primarily marseillevirus. Remarkably, some of these hits scored higher than those for Melbournevirus, supporting the presence of homologous proteins in these viruses (Supplementary Table 5).”

The ‘Discussion’ section now contains the following text:

“Additionally, the observation that the identified Jyvaskylavirus minor capsid protein sequences are shared across other marseillaviruses supports their essential structural and stabilizing roles in these viruses.”

At the same time, we have modified the ‘Materials and Methods’ section to include a reference to Supplementary Figure 5, where the use of ModelAngelo is mentioned. Additionally, a new Supplementary Figure 10 has been included to clarify how the residues built into the Melbournevirus density using ModelAngelo (without prior knowledge of any sequence) are subsequently matched with the Jyvaskylavirus sequences.

(11) Based on the author's statement, Iker Arriaga did all the cryoEM experiments. It is strange to me they are not placed higher on the author's list.

We thank you for this observation and agree with your comment. This manuscript has been in preparation for a few years, and the first draft had the author order defined before the structural data collection and analyses were completed. Iker participation was indeed important and substantial from the first draft to the submitted version and he definitely deserves a better author placement. We have modified the author order to accommodate this. Note that only the author order changed and that no author has been included or removed.